# Optical characterization of pure pollen types using a multi-wavelength Raman polarization lidar

Xiaoxia Shang[1], Elina Giannakaki[1,2], Stephanie Bohlmann[1,3], Maria Filioglou[1], Annika Saarto[4], Antti Ruuskanen[1], Ari Leskinen[1,3], Sami Romakkaniemi[1], and Mika Komppula[1]

[1]Finnish Meteorological Institute, Atmospheric Research Centre of Eastern Finland, 70211, Kuopio, Finland.
[2]Department of Environmental Physics and Meteorology, University of Athens, Athens, Greece.
[3]Department of Applied Physics, University of Eastern Finland, 70211, Kuopio, Finland
[4]Aerobiology Unit, University of Turku, Finland.

*Correspondence to*: Xiaoxia Shang (xiaoxia.shang@fmi.fi)

**Abstract.** We present a novel algorithm for characterizing the optical properties of pure pollen particles, based on the depolarization ratio values obtained in lidar measurements. The algorithm was first tested and validated through a simulator, and then applied to the lidar observations during a four-month pollen campaign from May to August 2016 at the European Aerosol Research Lidar Network (EARLINET) station in Kuopio (62°44'N, 27°33'E), in Eastern Finland. Twenty types of pollen were observed and identified from concurrent measurements with Burkard sampler; Birch (*Betula*), pine (*Pinus*), spruce (*Picea*) and nettle (*Urtica*) pollen were most abundant, contributing more than 90 % of total pollen load, regarding number concentrations. Mean values of lidar-derived optical properties in the pollen layer were retrieved for four intense pollination periods (IPPs). Lidar ratios at both 355 and 532 nm ranged from 55 to 70 sr for all pollen types, without significant wavelength-dependence. Enhanced depolarization ratio was found when there were pollen grains in the atmosphere, and even higher depolarization ratio (with mean values of 0.25 or 0.14) was observed with presence of the more non-spherical spruce or pine pollen. Under the assumption that backscatter-related Ångström exponent between 355 and 532 nm should be zero for pure pollen, the depolarization ratio at 532 nm of pure pollen particles was assessed, resulting to $0.24 \pm 0.01$ and $0.36 \pm 0.01$ for birch and pine pollen, respectively. Pollen optical properties at 1064 nm and 355 nm were also estimated. The backscatter-related Ångström exponent between 532 and 1064 nm was assessed as ~0.8 (~0.5) for pure birch (pine) pollen, thus the longer wavelength would be better choice to trace pollen in the air. The pollen depolarization ratio at 355 nm of 0.17 and 0.30 were found for birch and pine pollen, respectively. The depolarization values show a wavelength dependence for pollen. This can be the key parameter for pollen detection and characterization.

## 1   Introduction

Pollen has various effects on human health and the environment. The number of people suffering from allergies due to pollen inhalation is rising (Schmidt, 2016). Airborne pollen is recognized as one of the major agents of allergy-related diseases such as asthma, rhinitis, and atopic eczema (Bousquet et al., 2008). Pollen is also biogenic air pollutant which affects both the solar

radiation reaching the Earth and cloud optical properties by acting as seed for both cloud droplets and ice crystals (Steiner et al., 2015).

Various networks are built to monitor pollen concentrations at ground level using in situ instruments (Giesecke et al., 2010). In 2020, there is more than 1000 active pollen monitoring stations in the world (Buters et al., 2018,
https://oteros.shinyapps.io/pollen_map/, last access: 7 April 2020), with majority based on the Hirst principle (Hirst, 1952). Conventional method of pollen classification is based on pollen morphological characters using microscopy (Holt and Bennett, 2014; Weber, 1998). However, it requires complex procedures for the complete classification and identification, and the results are not publicly available online. Besides, pollen grains can be agile and change their visual nature before the analysis, e.g. undergo an osmotic shock (Miguel et al., 2006), which lead to errors in pollen characterization. Several studies on the long
distance transport of pollen (Rousseau et al., 2008; Skjøth et al., 2007; Szczepanek et al., 2017) have shown that pollen grains can be lifted up to several kilometers and be dispersed by wind over thousands of kilometers.

An increasing interest in pollen has arisen in the aerosol lidar community (Noh et al., 2013; Sicard et al., 2016). In our previous study (Bohlmann et al., 2019) we showed on the basis of an 11-day birch pollination period that lidar measurements can detect the presence of pollen grains in the atmosphere, and that the non-spherical pollen grains can generate strong depolarization
(we found a mean depolarization ratio of 0.26 for the birch-spruce pollen mixture). Therefore, it is possible to observe airborne pollen grains in the atmosphere using depolarization ratio in the absence of other depolarizing non-spherical particles (e.g. dust). We have also reported that lidar derived parameters (e.g. depolarization ratio and Ångström exponent) provide the possibility to identify different pollen types (e.g. birch and spruce pollen). However, the optical properties of pure pollen are still missing due to the fact that the atmospheric aerosol population is always a mixture of several particle types. For instance,
the depolarization ratio of pure pollen is an essential parameter needed to separate pollen backscatter from the background aerosol backscatter. Ångström exponent and lidar ratio, which are often used for aerosol typing, are also crucial parameters to be defined for pure pollen particles.

In this study, we present a novel method for characterizing the optical properties of pure pollen particles, based on a four months campaign. In Sect. 2, we introduce the pollen campaign and the instruments. In Sect. 3, we present the methodology
and describe a novel algorithm to estimate the depolarization ratio value for pure pollen. This algorithm is tested and validated through a simulator. In Sect.4, we report the results: Firstly, the pollen information observed by the Burkard sampler and lidar retrieved optical properties for the pollen layer are presented. Secondly, the novel algorithm of Sect.3 is applied to the lidar observations in Sect. 4.3 to retrieve the optical properties for pure pollen. Section 5 is devoted to the summary and conclusion.

## 2   Site and instruments

The measurement campaign was performed from May to August 2016, at the Kuopio station of the European Aerosol Research Lidar Network (EARLINET) in Vehmasmäki (62°44'N, 27°33'E, elevation of 190 m above sea level). This rural site is mainly surrounded by forest, located ~18 km from the city center of Kuopio, in Eastern Finland. Finland provides suitable conditions

for the observation of pollen as 78 % of Finland's total area is covered by forests. Airborne *Betula* spp. (birch) pollen is one of the most recognized aeroallergens in northern European countries and the most important cause of pollen allergy (Sofiev et al., 2015; Yli-Panula et al., 2009). The predominant *Betula* species include *B. pendula* and *B. pubescens*, while *B. nana* and *B. pubescens subsp. czerepanovii* can be found in northern parts of the country. As to conifers, *Pinus sylvestris* and *Picea abies*

are the most prevalent and *P. sylvestris* pollen typically causes the highest peaks during the pollen season. *P. sylvestris* and *P. abies* are the only naturally growing species of their genre in Finland. Compared to many other European countries, relatively clean background atmospheric conditions in Finland favour pollen detection and further separation of contributions of pollen backscattering from total scattering by using lidars, since there are less other particles, particularly dust, which would complicate the analysis.

The Kuopio station is operated by the Finnish Meteorological Institute, and it is equipped with a ground-based multi-wavelength Raman polarization lidar Polly[XT] (Engelmann et al., 2016), Doppler lidar, and in-situ instruments next to a 318 m mast (for the meteorological observations) since autumn 2012 (Hirsikko et al., 2014). Polly[XT] has three emission wavelengths (355, 532 and 1064 nm) and seven detection channels (including three emitted wavelengths channels, three inelastic Raman-shifted wavelengths channels (387, 407 and 607 nm) and the cross-polarization channel at 532 nm). Polly[XT] has an initial

spatial resolution of 30 m and a temporal resolution of 30 s. During daytime, the Klett-Fernald method (Fernald, 1984; Klett, 1981) is applied using the elastic signals to retrieve the extinction coefficient which describes the combined effect of particle absorption and scattering, and the backscatter coefficient which describes particle backscattering at 180° scattering angle. During night-time, profiles of extinction and backscatter coefficients at 355 and 532 nm can be derived independently using elastic and inelastic Raman-shifted wavelengths (387 and 607 nm), based on the Raman inversion (Ansmann et al., 1992). The

ratio of extinction to backscatter coefficient is called lidar ratio (LR), which is considered an important parameter to separate particle types, as it depends on their single scattering albedo and backscatter phase function, thus being a function of size distribution and chemical composition. The cross- and total- polarization channels of the Polly[XT] allow the retrieval of the volume linear depolarization ratio (VDR) and particle linear depolarization ratio (PDR) at 532 nm, which provide information on the shape of the scattering particles. Multi-wavelength measurements (355 nm, 532 nm and 1064 nm) enable the

determination of Ångström exponents between each wavelength pairs, which are related to the particle nature, mostly the size. Previous studies show (e.g. Eck et al., 1999) that Ångström exponent values greater than 2 indicate small particles associated with combustion byproducts, whereas Ångström exponent values less than 1 indicate large particles like sea salt and dust.

In addition to the lidar measurements, a Hirst-type Burkard pollen sampler (Hirst, 1952) was placed 4 meters above ground level (agl) next to the lidar instrument. The Burkard sampler enables identification of pollen types and concentration

microscopically with a 2-hour time resolution. More detailed descriptions of the pollen sampler and Polly[XT] used during this campaign can be found in Bohlmann et al. (2019) and reference therein.

## 3    Methodology – a synthetic simulator

In this study, we provide a novel method and develop an algorithm to estimate the depolarization ratio value for pure pollen particles. This algorithm is first tested through a simulator (Sect.3) using the synthetic lidar data, and then applied to the real lidar observations (Sect.4.3). The simulator includes a direct model and an inverse model modules (the block diagram is shown

in Fig. S1 in the supplement); Similar ones have already been used for forest and aerosol studies (Shang et al., 2018; Shang and Chazette, 2015). Synthetic data are used in this section to present our methodology. We mainly consider two wavelengths: $\lambda_1 = 355$ nm and $\lambda_2 = 532$ nm, another wavelengths combination of 532 and 1064 nm will be briefly discussed at the end of Sect. 3.1.

### 3.1    Direct model – generation of synthetic optical profiles

Two aerosol populations, pollen (depolarizing) and background (non-depolarizing) aerosols, are considered in this simulation. The optical and physical parameters used in the direct calculation are presented in Table 1; these parameters are named as "initial values" for the simulation. The values are based on our lidar measurements (Bohlmann et al., 2019) or literature (e.g. Illingworth et al., 2015). The *background* here refers to non-depolarizing background aerosols (non-pollen particles), which can be polluted continental or biomass burning aerosols. The depolarization ratio at both 355 and 532 nm of non-pollen particle

($\delta_{\text{background}}$) are selected as 0.03, which is a mean value for pollen-free periods at our measurement site. Bohlmann et al. (2019) shows that the pollen can generate strong depolarization, thus the depolarization ratio at 532 nm of pure pollen particle ($\delta_{\text{pollen}}$) are selected as 0.35 as the initial value for the simulation in this section. Pollen grains are quite big and thus can be assumed to be wavelength independent on the backscatter at wavelengths of 355 nm and 532 nm, with the backscatter-related Ångström exponent ($\text{Å}_{\text{pollen}}$) of 0. The backscatter-related Ångström exponent between 355 and 532 nm of non-pollen particle

($\text{Å}_{\text{background}}$) is assumed to be 2, regarding the previous studies over Arctic regions (e.g. Schmeisser et al., 2018; Tomasi et al., 2012). Note that these values can be changed freely for the simulation under 2 constraints: i. depolarization ratio of pollen (depolarizing one) should be higher than the depolarization ratio of background aerosol (non-depolarizing one), ii. the values of backscatter-related Ångström exponent for pollen and non-pollen particle should be different. In addition, the conclusion of the simulation section is not depended on the assumed profile shape or height; and the initial values are not critical for

presenting the overall approach.

The extinction coefficient profiles of these two aerosol layers are assumed to following a Gaussian distribution. The optical depth (OD) of the input background aerosol layer is fixed to be 0.1 in this simulation. In order to simulate different pollen contribution to the total aerosol load, we change the pollen load by selecting different input values for the pollen layer OD. Pollen OD is used as 0.002, 0.01, 0.02, 0.05, 0.1, and 1, thus six pollen backscatter coefficient profiles are simulated. One

example of simulated pollen and background backscatter coefficients is shown in the supplement (Fig. S2a) for pollen OD of 0.1. The pollen layer is defined as the layers below 1 km.

Next, pollen layer and background layer are summed up (Eq.1), and then the vertical profiles of aerosol backscatter coefficient, lidar ratio and Ångström exponent of the total particles are simulated (e.g., Fig. S2b). The Ångström exponent describes the wavelength-dependence on aerosol optical properties (Ångström, 1964). Backscatter-related Ångström exponent between two wavelengths of $\lambda_1$ and $\lambda_2$ (denoted as Å) can be expressed as Eq.2.

$$\beta_{\text{particle}}(\lambda, z) = \beta_{\text{pollen}}(\lambda, z) + \beta_{\text{background}}(\lambda, z) \tag{1}$$

$$\text{Å}_x(\lambda_1, \lambda_2, z) = -\frac{\ln\left(\frac{\beta_x(\lambda_1, z)}{\beta_x(\lambda_2, z)}\right)}{\ln\left(\frac{\lambda_1}{\lambda_2}\right)} \tag{2}$$

The index *x=pollen, background* or *particle* denotes the backscatter-related Ångström exponent of pollen, background or total particles.

Vertical profiles of particle linear depolarization ratio (PDR, denoted as $\delta_{\text{particle}}$) can be also calculated following Eq.3 (the detailed calculation is given in the supplement).

$$\delta_{\text{particle}} = \frac{\frac{\beta_{\text{pollen}} * \delta_{\text{pollen}}}{\delta_{\text{pollen}} + 1} + \frac{\beta_{\text{background}} * \delta_{\text{background}}}{\delta_{\text{background}} + 1}}{\frac{\beta_{\text{pollen}}}{\delta_{\text{pollen}} + 1} + \frac{\beta_{\text{background}}}{\delta_{\text{background}} + 1}} \tag{3}$$

Theoretically, these parameters can be derived directly from lidar observations. In order to keep the consistency of the availability of lidar-derived parameters, particle backscatter coefficient at 532 nm, PDR at 532 nm, and backscatter-related Ångström exponent between 355 and 532 nm simulated for these 6 cases (shown in Fig.1) will be used later as input of inverse model.

Pollen backscatter contribution, denoted as $\chi_{\text{pollen}}$ (Eq.4), is defined as the ratio of pollen backscatter coefficient ($\beta_{\text{pollen}}$) and the total particle backscatter coefficient ($\beta_{\text{particle}}$). Note that the use of "particle" here is to distinguish from "molecular".

$$\chi_{\text{pollen}}(\lambda, z) = \frac{\beta_{\text{pollen}}(\lambda, z)}{\beta_{\text{particle}}(\lambda, z)} \tag{4}$$

We investigate here the relationship of the backscatter-related Ångström exponent of total particles ($\text{Å}_{\text{particle}}$) and pollen backscatter contribution ($\chi_{\text{pollen}}$) at different wavelengths (the detailed calculation is given in the supplement), resulting a power law relationship:

$$\left(\frac{\lambda_1}{\lambda_2}\right)^{-\text{Å}_{\text{particle}}(\lambda_1,\lambda_2)} = \left(\left(\frac{\lambda_1}{\lambda_2}\right)^{-\text{Å}_{\text{pollen}}(\lambda_1,\lambda_2)} - \left(\frac{\lambda_1}{\lambda_2}\right)^{-\text{Å}_{\text{background}}(\lambda_1,\lambda_2)}\right) \cdot \chi_{\text{pollen}}(\lambda_2) + \left(\frac{\lambda_1}{\lambda_2}\right)^{-\text{Å}_{\text{background}}(\lambda_1,\lambda_2)} \tag{5}$$

The wavelength pairs $(\lambda_1, \lambda_2)$ are selected as (355,532), (532,355), or (1064,532) in this study. In order to simplify the calculation, we introduce two parameters η, and η′ as a function of the backscatter-related Ångström exponent between 355 and 532 nm or between 532 and 1064 nm, for the total particle backscatter coefficients:

$$\begin{cases} \eta = \left(\frac{355}{532}\right)^{-\text{Å}_{\text{particle}}(355,532)} \\ \eta' = \left(\frac{1064}{532}\right)^{-\text{Å}_{\text{particle}}(1064,532)} \end{cases} \tag{6}$$

The pairs of parameter η or η′ and $\chi_{\text{pollen}}$ at different wavelengths resulting linear relationships are reported in Table 2. For example, the pollen backscatter contribution at 532 nm ($\chi_{\text{pollen}}(532)$) is inversely proportional to the parameter η. Using the previous 6 simulated cases, a perfect linear relationship is found to fit the η versus $\chi_{\text{pollen}}(532)$ (Fig.2).

### 3.2 Inverse model – retrieval of depolarization ratio

In this section, we present the inverse model to retrieve the depolarization ratio of pure pollen particles. Tesche et al. (2009) provide a method to separate dust and non-dust contributions, based on the difference of the depolarization ratio values of these two types. This separation method is applied here to separate the 2 simulated aerosol types.

The pollen backscatter coefficient can be separated from the total particle backscatter coefficient (calculated from Eq.3), expressed as:

$$\beta_{\text{pollen}} = \beta_{\text{particle}} \frac{(\delta_{\text{particle}} - \delta_{\text{background}})(1 + \delta_{\text{pollen}})}{(\delta_{\text{pollen}} - \delta_{\text{background}})(1 + \delta_{\text{particle}})} \qquad (7)$$

The only remaining unknown to solve the Eq.7 is the depolarization ratio for pure pollen ($\delta_{\text{pollen}}$). Next we use previously simulated $\beta_{\text{particle}}$ and $\delta_{\text{particle}}$, and the assumed $\delta_{\text{background}}$. From now on, 532 nm will be the default wavelength (if not otherwise specified). The wavelength pair ($\lambda_1, \lambda_2$) is selected as (355,532) in this section. Mean values of optical properties inside the pollen layer are considered in this study; it is also possible to use values of each bin of the synthetic profile which

will lead to the same conclusion. Mean values of backscatter-related Ångström exponent between 355 and 532 nm inside the pollen layer, denoted as Å(355,532), can be easily retrieved.

Mathematically, the depolarization ratio for pure pollen can be calculated using Eqs.4,5,7, as other variables are known or can be assumed. Nevertheless, we developed a retrieval method for this inverse model, so that it can be easier applied to the real lidar measurements, especially for investigating the depolarization ratio with different values of the unknown $Å_{\text{pollen}}$. An

20 iterate approach is used. In the first step, the depolarization ratio for pure pollen was assumed to be several different values (within the range between 0.03 to 1), denoted as $\delta_x$, in the simulator. Related pollen backscatter contribution ($\chi_{\text{pollen}}(532)$) inside the pollen layer, can be retrieved using Eqs.4 and 7. As its value depends on the assumed pollen depolarization ratio ($\delta_x$), it can be expressed as $\chi_{\text{pollen}}(\delta_x, 532)$.

The relationship of Å(355,532) and $\chi_{\text{pollen}}(\delta_x, 532)$ was investigated using the parameter η (Eqs.5 and 6). Examples of

25 scatter plots using mean values of η and $\chi_{\text{pollen}}(\delta_x, 532)$ in the pollen layer for cases under the assumptions of $\delta_x$=0.1, 0.2, 0.3, 0.4 and 0.5 are shown in Fig.3. For these relationships, perfect linear fits (linear regression relationship) can be found and plotted as dotted lines in the Fig.3, following the simplified equation from Eqs.5 and 6:

$$\eta\big(\chi_{\text{pollen}}(\delta_x, 532)\big) = a_1 \cdot \chi_{\text{pollen}}(\delta_x, 532) + a_0 \qquad (8)$$

The fitting coefficient ($a_1, a_0$) values to determine the estimated parameter η are defined as in Eq.5. Until this step of the

30 inverse model, no assumption on the $Å_{\text{pollen}}$ was made, thus $a_1$ varies for different assumed values of $\delta_x$. But $a_0$ is constant as the $Å_{\text{background}}$ is known. Theoretically, for each linear fit equation, $\chi_{\text{pollen}}(\delta_x, 532)$ values can range from 0 to 1, with 0

meaning no pollen and 1 meaning 100 % pollen in the observed aerosol particle population. Therefore, for each assumed $\delta_x$, the η value for $\chi_{\text{pollen}}(\delta_x, 532)=1$ can be defined as the value for the pure pollen, and denote as $\eta_{\text{pure}}(\delta_x, 532)$.

In Sect. 3.1, the initial value of the backscatter-related Ångström exponent between 355 and 532 nm of pure pollen (denoted as $\mathring{A}_{\text{pollen}}$) is 0, which results in an initial value of 1 for the parameter η. In this simulation, we assumed that the same value ($\widehat{\mathring{A}}_{\text{pollen}}=0$) should be retrieved; the goal was thus to find the value of 1 for $\eta_{\text{pure}}$. From previous results shown in Fig.3, we can see a $\delta_x$ between 0.3 to 0.4 may result in a $\eta_{\text{pure}}=1$ (the black triangle in Fig.3).

Hence, in the second step, more $\delta_x$ values between that range (0.3 – 0.4) were used in the simulation, and one can retrieve the relative value of $\eta_{\text{pure}}(\delta_x, 532)$ for each case. These values are presented in Fig.4. The relationship between $\delta_x$ and $\eta_{\text{pure}}(\delta_x, 532)$ is not perfectly linear, but for these data inside the considered range, a good linear fit can be found with high correlation coefficients ~-1. As there is noise in real lidar measured profiles, two or more values of $\delta_x$ may be found as good solutions. However, after we introduce this additional second linear fit, only one solution will be retrieved in the end.

Finally, under the assumption of $\widehat{\mathring{A}}_{\text{pollen}}=0$, pollen depolarization ratio of 0.35 was found, resulting in a $\eta_{\text{pure}}=1$ (shown by the black triangle in Fig.4). This result is exactly the same as the initial value of the direct model, which validates the algorithm and provides the feasibility of using this inverse model to retrieve the pure pollen depolarization ratio values. A detailed flow chart of this inverse model is given in Fig.5. Note that the values of $\delta_x$ can be chosen freely, for values bigger than background depolarization ratio and smaller than 1. This method can also be applied to other two aerosol types (e.g., dust and non-dust aerosols), under the condition that the depolarization ratio of one aerosol type is the only unknown parameter, and other parameters are known or can be assumed, as long as both the depolarization ratio and the backscatter-related Ångström exponent of the two aerosol types are different.

## 3.3 Uncertainty study

The uncertainty study of this method is investigated in this section. The input parameters (i.e. initial values) of the direct model are defined in Sect.3.1, with optical depth (OD) of the background aerosol of 0.1, and pollen OD of 0.002, 0.01, 0.02, 0.05, 0.1, or 1. Nonetheless, some input parameters (e.g., the pollen depolarization ratio $\delta_{\text{pollen}}$ and the backscatter-related Ångström exponent for pollen $\mathring{A}_{\text{pollen}}$) were selected as different initial values for different uncertainty studies, which are clarified in each paragraph. The output of each direct model simulation were then used as the input of the inverse model.

Under the ideal condition, which means there is no noise on the input profiles for the inverse model, the depolarization ratio of pollen (depolarizing one) can be retrieved perfectly as long as the value is higher than the depolarization ratio of background aerosol (non- depolarizing one). $\delta_{\text{pollen}}$ of 0.04 has been tested, and the correct value was successful retrieved. Note that for this case, the assumed values of $\delta_x$ should be selected as lower values (e.g. from 0.03). The more values of $\delta_x$ used in the inverse model, the better precision will be for the results, but also longer computation time is needed. It is also possible to

combine the first and second steps of inverse model, by using many assumed values of $\delta_x$ (e.g. 0.032, 0.033, 0.034, …, 0.98, 0.99) for the first step, at the cost of long computation time.

In the presented cases, we assumed that the backscatter-related Ångström exponent between 355 and 532 nm of pure pollen to be used in the inverse model (denoted as $\widehat{\text{Å}}_{\text{pollen}}$) is 0, which was the same as the initial value ($\text{Å}_{\text{pollen}}$) of direct model. But in the reality, such information is not always available. Under different initial values of $\text{Å}_{\text{pollen}}$, there will be a bias on the estimated values of pollen depolarization ratio if the assumed value is different (i.e. $\widehat{\text{Å}}_{\text{pollen}} \neq \text{Å}_{\text{pollen}}$). For example, if the initial value $\text{Å}_{\text{pollen}}$ is 0.25 (i.e. $\eta_{\text{pure}}$=1.11), but we keep the assumption of $\widehat{\text{Å}}_{\text{pollen}}$=0 in the inverse model, the estimated pollen depolarization ratio is found to be 0.39 with a bias of 0.04 (show in Fig. S3 in the supplement). The uncertainty due to the difference between the initial value of $\text{Å}_{\text{pollen}}$ and assumed $\widehat{\text{Å}}_{\text{pollen}}$ were simulated (show in Fig. S4 in the supplement), where $\widehat{\text{Å}}_{\text{pollen}}$ is always assumed as 0 in the inverse model. For initial values of $\text{Å}_{\text{pollen}}$=±0.5 (i.e. bias of 0.5 on the assumed value of 0), relative uncertainties were assessed as ~30 %. This uncertainty due to the difference of initial values of $\text{Å}_{\text{pollen}}$ and $\text{Å}_{\text{background}}$ was also investigated. The larger the difference between two values ($\text{Å}_{\text{background}} - \text{Å}_{\text{pollen}}$), the smaller the uncertainty. For instance, if we use 3 (instead of 2) as the initial value of $\text{Å}_{\text{background}}$, the estimated pollen depolarization ratio is 0.37 (instead of 0.39) with a smaller bias for the above example.

Further on, we investigate the random uncertainty due to the noise on input lidar profiles, using the simulator based on a Monte Carlo approach. The parameters for the 6 cases simulated earlier (as defined in Sect. 3.1, with values given in Table 1) are used again in this simulation, but noises are additionally added, considering normal statistical distributions, which are introduced by a normal random generator (Fig. S1). The PDR and Å are calculated from particle backscatter coefficients, so we only need to apply different noise levels to the particle backscatter coefficients in the direct model, and related PDR and Å with noise can be retrieved. To simplify the problem, the initial noise levels for both backscatter coefficients at 355 and 532 nm were considered under the same assumptions. We defined "1 group" as 1 draw of 6 simulated backscatter profiles with a certain noise level; these 6 backscatter profiles are with pollen OD of 0.002, 0.01, 0.02, 0.05, 0.1, and 1. For each statistical simulation, we used 200 draws (i.e. 200 groups of profiles). This uncertainty study was investigated by 2 parts:

**i. Fix input pollen depolarization ratio, and change noise levels**. We used 0.35 as the initial pollen depolarization ratio. In case of taking 10 % as the noise level on the backscatter coefficients, one group of 6 simulated profiles with noise are shown in Fig.6. Pollen depolarization ratio of 0.354 was found for this group using the inverse model, with a bias of 0.004 compared to the initial value of 0.35. Similarly, pollen depolarization ratio values were retrieved for each of the 200 generated groups. These 200 values had a mean value of 0.351 ± 0.009, thus an uncertainty of 0.009 (relative uncertainty of 2.6 %) was found. We changed the noise levels (e.g., 1 %, 10 %, 20 %, 40 %, and 60 %) on the backscatter coefficients by the normal random generator, and 200 draws were performed for each statistical simulation under each noise level. The uncertainties of the retrieved pollen depolarization ratio against the noise levels were assessed and shown in Fig.7a.

**ii. Fix noise level and change input pollen depolarization ratio**. In the second simulation, we keep 10 % as the noise level on the backscatter coefficients, and change the input pollen depolarization ratio values as 0.1, 0.2, 0.3, 0.4, and 0.5. Under each assumption, 200 draw were performed to derive the uncertainties values, which are reported in Fig.7b. Relative uncertainties on retrieved pollen depolarization ratio of 1.6 % to 2.8 % were found.

From simulation results, small uncertainty and good accuracy were found using this algorithm. Nevertheless, even with the introduced noise levels, these simulations were still performed under quasi ideal condition. For each simulated group, 6 cases were used to provide a wide range of values of $\chi_{pollen}$ (from ~0.05 to ~0.95), which leading good constraints to find a fitting line for the regression relationship of $\chi_{pollen}$ and $\eta$ (Eq.6) (e.g. Fig.3). If only 3 cases (with Pollen OD of 0.01, 0.02, and 0.05) were used for each group, 2 to 5 times bigger uncertainties were found. It is hard to give qualitative values for such uncertainty study, but the wider range of $\chi_{pollen}$ values are in the data set, the better the retrievals will be. The vertical resolution used here was 30 m (as the raw resolution of our lidar); and increasing the vertical resolution of the lidar would result in smaller uncertainty in simulation.

## 4    Results

### 4.1    Pollen grain and intense pollination period

During the four months campaign, 20 pollen types were observed and identified from the samples collected with the Burkard sampler. Six from broadleaved trees, observed from end of April to mid of June; three from coniferous trees, with pollination period from mid of May to mid of June; and eleven from grass/weed, observed mainly in July and August. Among them, birch (*Betula*), pine (*Pinus*), spruce (*Picea*) and nettle (*Urtica*) pollen were most abundant, contributing to more than 90 % of the total pollen load, regarding number concentrations. The surrounding forest is mixed in terms of the tree species, but the pollination periods of different dominant pollen types are distinct, as can be seen from the Burkard observed number concentration of specific pollen types shown in Fig.8a.

Microphotographs of pollen grains for the dominant pollen types are shown in Fig.8b (photos taken from www.paldat.org, last access: 7 April 2020). Pine and Spruce pollen belong to *Pinaceae* family, which pollinate profusely and greatly contribute to the pollen counts. However, they are rarely considered as allergenic. Their pollen grains are large due to their sacs or bladders, which make them easy to identify. Among winged grains, the body is sub-spheroidal to broadly ellipsoidal. The longest axis (sacci included) of *Pinus sylvestris* (Scots pine) pollen grains is 65-80 µm, while in *Picea abies* (Norway spruce) the axis is longer, 90-110 µm (Nilsson et al., 1977). Birch pollen can cause severe pollinosis, and is recognized as one of the most important allergenic source (D'Amato et al., 2007). Birch pollen grains are sub-oblate to oblate. *B. pubescens* pollen grains are 18-24 × 22-28 µm in size (Nilsson et al., 1977) and *B. pendula* (Silver birch) pollen grains are more or less of the same size (spoken communication with Sanna Pätsi from Aerobiology, University of Turku). Nettle is considered moderately allergenic, both in terms of skin tests and amount of exposure to the pollen in the air. Nettle *(Urtica dioica)* pollen grains are

oblate-spheroidal to spheroidal, and are quite small with size of 13-17 × 15-20 µm (Nilsson et al., 1977). Information of the dominant pollen types are reported in Table 3, where the pollen season is defined using the 95 % method (Goldberg et al., 1988). The start of the season was defined as the date when 2.5 % of the seasonal cumulative pollen count was trapped and the end of the season when the cumulative pollen count reached 97.5 %.

Four intense pollination periods (IPPs) are defined considering the pollen seasons and the daily mean pollen concentration values of these 4 dominant pollen types (Table 3). A minimum value of 300 no.m$^{-3}$ (for daily mean pollen concentration) was used as the threshold for the determination of IPP-1 and IPP-3, whereas a smaller threshold of 20 no.m$^{-3}$ was used for IPP-2 and IPP-4. In addition, the availability of lidar measurements were considered for the IPP definition. IPP-1 and -2 are selected within the birch pollen season. During IPP-1, almost only birch pollen is observed (97 % contribution in number concentration),

while during IPP-2, spruce pollen is additionally present in the air with 14 % contribution. IPP-3 consists of 2 periods within the pine pollen season, separated by a few days with frequent low level clouds (below 1 km) or rain, causing the relatively low pine pollen concentration between these two periods. IPP-4 is defined for nettle pollen study for 3 separate short pollination periods in July and August.

### 4.2 Optical properties of pollen layer

#### 4.2.1 Pollen layer

A pollen layer in the lidar measurements is defined as the lowest observed layer. The layer boundaries are determined using the gradient method (Bösenberg and Matthias, 2003; Flamant et al., 1997; Mattis et al., 2008) based on lidar-derived

backscatter coefficient profile at 532 nm wavelength. More detailed description of the layer definition method is described in Bohlmann et al. (2019). Two-hour time averaged lidar profiles are used in this study to match the pollen sampler time resolution. The retrieved pollen layers are shown in Fig.9a. With an overlap correction applied in this study, the lower limit for reliable backscatter profiles was about 600 m agl. Statistical values of the pollen layer top height agl for the four IPPs were 1.5 ± 0.3 km, 1.3 ± 0.3 km, 1.3 ± 0.4 km, and 1.2 ± 0.3 km, respectively (Fig.9b). The lowest layer top height was found for

the nettle pollen, belonging to herbaceous species. For the relatively larger spruce and pine pollen, the layer top heights were lower compared to the smaller birch pollen.

#### 4.2.2 Lidar-derived optical properties

Mean values of lidar derived optical properties inside the detected pollen layers were retrieved (Table 4); these optical values represent the atmosphere with presence of pollen (thus the mixture of pollen with other aerosols).

Lidar ratio (LR) at 532 nm and LR at 355 nm for pollen layers were retrieved using the standard Raman method (Ansmann et al., 1990) during night-time measurements. The mean values are reported in Table 4, and boxplots of LR at 532 nm and ratio of LRs are shown in Fig.10 (a, b). Although the number of available profiles is limited, our results indicate that pollen are medium to high absorbing particles with values from 55 to 70 sr for all pollen types. For birch dominant IPP-1 and nettle

dominant IPP-4, LR of pollen layers at 532 nm is slightly larger than LR at 355 nm. This behaviour is reversed for IPP-3 (pine dominant) and IPP-2 (mixture of birch and spruce). However, no significant wavelength-dependence can be determined on LR values accounting the uncertainties.

The depolarization ratio was clearly enhanced when there were pollen grains in the air, and even higher depolarization ratios were observed with presence of the more non-spherical spruce and pine pollen. Lidar derived PDR values of detected pollen layers for the whole periods of each IPP are shown in Table 4 and Fig.10c. This indicates the depolarization ratio is the most proper indicator for pollen type. The extinction-related (not shown in this study) and backscatter-related Ångström exponent were also retrieved for pollen layers. The difference on the Ångström exponent for IPPs is much less evident, as the boxplot of backscatter-related Ångström exponent between 355 and 532 nm shows (Fig.10d). The use of Ångström exponent to characterize pollen is quite delicate, as its value depends a lot on the background aerosol. Nevertheless, a clear tendency to smaller Ångström exponent with increasing depolarization ratio can be found, as is reported in Bohlmann et al. (2019). Thus under same or similar background conditions, the Ångström exponent can be an indicator for pollen type. Even though we assumed that pollen grains were evenly distributed inside the pollen layer, bigger pollen contribution in the aerosol mixture near the ground was observed.

## 4.3 Estimation of optical properties for pure pollen from lidar observations

So far, we have retrieved the optical properties of the pollen layers, but the values for pure pollen are still unknown. In this section, the novel methodology presented in Sect.3 is applied to the real lidar observations to estimate the optical properties for pure pollen particles.

### 4.3.1    Pollen optical properties at 532 nm

The method given in the inverse model module was applied to the real lidar observations in this section to retrieve the depolarization ratio at 532 nm for pure pollen. We assume that there are only pollen and non-depolarizing background aerosols in the air, which is reasonable because of the clean aerosol conditions at the measurement site.

For the first step, the depolarization ratio at 532 nm of pure pollen ($\delta_x$) was assumed to be 0.2, 0.3, 0.4, or 0.5, and the depolarization ratio at 532 nm of non-pollen particles ($\delta_{background}$) was assumed to be 0.03. Under each assumption, we calculated the pollen backscatter coefficient during every IPPs, and thus extract the related pollen backscatter contribution inside the pollen layer ($\chi_{pollen}(\delta_x, 532)$). Mean values of backscatter-related Ångström exponents between 355 and 532 nm inside the pollen layer were retrieved and denoted as Å(355,532). The relationship of Å(355,532) and $\chi_{pollen}(\delta_x, 532)$ of pollen layers in each IPP was investigated using the parameter η (Eq.6). The scatter plots using mean η and $\chi_{pollen}(\delta_x, 532)$ under different values of assumed $\delta_x$ (0.2, 0.3, 0.4, or 0.5) for IPP-1 and IPP-3 are given in the supplement (Fig. S5 for IPP-1 and in Fig. S6 for IPP-3).

Based on results from the first step, in the second step, more $\delta_x$ values between 0.2 to 0.3 for IPP-1 (between 0.3 to 0.4 for IPP-3) were used for the calculations. Linear fitting lines were generated for the η and $\chi_{\text{pollen}}(\delta_x, 532)$ (Eq.8) under each assumed $\delta_x$. For these fitting lines, the η value for $\chi_{\text{pollen}}(\delta_x, 532) = 1$ was retrieved, denoted as $\eta_{\text{pure}}(\delta_x, 532)$ and reported in Fig. 11. $\eta_{\text{pure}}$ presents the η values when the pollen contribution in the observed aerosol particle population is 100 %. Using these estimated $\eta_{\text{pure}}(\delta_x, 532)$ and $\delta_x$, linear fits (shown by dotted lines in Fig. 11) can be assessed with high correlations. Further on, $\delta_x$ value which results in a certain value of $\eta_{\text{pure}}(\delta_x, 532)$ could be assumed as the depolarization ratio value of pure pollen. Under the assumption that the backscatter-related Ångström exponent between 355 and 532 nm of pure pollen (denoted as $\text{Å}_{\text{pollen}}$) is 0 (i.e. $\eta_{\text{pure}}=1$), depolarization ratio of 0.24 or 0.36 were found for IPP-1 or IPP-3, respectively, which are related to the pure birch or pure pine pollen (Table 5). The scatter plots of mean η and $\chi_{\text{pollen}}(\delta_x, 532)$ are shown in Fig. 12: (a) for IPP-1 with the pollen depolarization ratio of 0.24, and (b) for IPP-3 with the pollen depolarization ratio of 0.36. Good linear regression relationships are found for both cases, and two things should be highlighted: (1) $\text{Å}_{\text{pollen}}$ is 0 (i.e. $\eta_{\text{pure}}=1$) for 100 % pollen in the observed aerosol particle population (i.e. $\chi_{\text{pollen}}=1$); (2) without pollen in the air (i.e. $\chi_{\text{pollen}}=0$), the backscatter-related Ångström exponent between 355 and 532 nm of non-pollen particles ($\text{Å}_{\text{background}}$) can be calculated, resulting values of 2.0 for IPP-1 and 1.9 for IPP-3 (i.e. η of 2.28 for IPP-1, 2.18 for IPP-3). There is no values of Ångström exponent for pure pollen in the literature, but this assumption ($\text{Å}_{\text{pollen}}= 0$) is almost realistic, as pollen grains are quite big, and thus can be assumed to be wavelength independent on the backscatter at wavelengths of 355 nm and 532 nm. For big particles as dust, Mamouri and Ansmann (2014) reported extinction-related Ångström exponent between 440 and 675 nm with values of -0.2 for coarse dust and 0.25 for total dust.

Uncertainty study was investigated based on method describe in Sect.3.3 using a Monte Carlo approach. The overall relative uncertainties of the lidar-derived backscatter coefficients are of the order of 5 %–10 % (Baars et al., 2012), we took 10 % here in the simulation. Initial pollen depolarization ratio values were selected as 0.24 for birch and 0.36 for pine for the uncertainty simulation; initial backscatter-related Ångström exponent between 355 and 532 nm of non-pollen particles were selected as 2.0 and 1.9 for IPP-1 and IPP-3, respectively. Based on the lidar observations (Fig. 12), the simulated cases were selected so that the $\chi_{\text{pollen}}$ values range from 2 % to 60 % for birch and 2 % to 90 % for pine. The initial input $\text{Å}_{\text{pollen}}$ in the direct model and assumed $\hat{\text{Å}}_{\text{pollen}}$ in the inverse mode were both selected as 0. Estimated uncertainties were found as 2.4 % for birch and 2.9 % for pine (Table 5). Note that the different initial input values of $\text{Å}_{\text{pollen}}$ may introduce important additional bias. If we assume the true value of $\text{Å}_{\text{pollen}}$ is between -0.5 to 0.5 (i.e. values of $\eta_{\text{pure}}$ from 0.82 to 1.22, shown by red dotted lines in Fig. 11), depolarization ratios of 0.19 to 0.27 can be found for birch pollen, and 0.26 to 0.44 can be found for pine pollen. The optical properties of pure pollen is lacking in the literature. Cao et al. (2010) measured the linear depolarization ratio of different pollen types in an aerosol chamber, by disseminating 2 g of the selected pollen; They determined a linear depolarization ratio at 532 nm for paper birch of 0.33, and for Virginia pine of 0.41. These values are higher than what we

retrieved in this study, but it has to be kept in mind that these two experiments have been conducted in quite different environments and conditions.

The retrieval of depolarization ratios for pure spruce or pure nettle pollen was not possible with this dataset. During IPP-2, there was always a mixture of birch and spruce pollen with variable mixing rate; in addition, the number of available measurements is limited. For nettle pollen, we have observed relatively small depolarization ratio values, together with a small variation, which makes the separation more challenging.

### 4.3.2    Pollen optical properties at 1064 nm and 355 nm

Similar study was performed to investigate the relationship between backscatter-related Ångström exponent between 532 and 1064 nm (Å(1064,532)) and pollen backscatter contribution at 532 nm, here we use another parameter $\eta'$ (Eq.6), which is a function of Å(1064,532), for the total particle backscatter. From the earlier simulations, we found out that the pollen backscatter contribution at 532 nm ($\chi_{pollen}(532)$) is proportional to the parameter $\eta'$, considering the Eq.5 using the wavelength pair of $\lambda_1$=1064 and $\lambda_2$=532.

The inverse model was applied for several assumed pollen depolarization ratios at 532 nm (ranging from 0.2 to 0.6), and no values of $\eta'$=1 (i.e. Å(1064,532)$_{pollen}$=0) was found (Fig. S5, S6, S7 in the supplement). This result may due to the fact that the laser beam at longer wavelengths would be more sensitive to bigger particles (pollen). Thus, there is some wavelength dependence on the backscattering between 532 and 1064 nm. The backscatter-related Ångström exponent between 532 and 1064 nm of non-pollen particles, denoted as Å$_{background}$(532,1064), can be calculated using Eq.5 and the fitting equations in Fig S5b and S6b, considering no pollen in the air (i.e. $\chi_{pollen}$=0). Å$_{background}$(532,1064) value of 1.0 or 1.1 (i.e. $\eta'$= 0.50 or 0.46) was estimated for IPP-1 or IPP-3, respectively. Considering the previously estimated depolarization ratios at 532 nm for pure birch (pine) pollen of 0.24 (0.36), the related $\eta'$ was found to be 0.58 (0.69), corresponding to the value of ~0.8 (~0.5) for the backscatter-related Ångström exponent between 532 and 1064 nm. Extinction-related Ångström exponent is characterized mainly by the particle size, whereas the backscatter-related Ångström exponent depends on both the particle size and refractive index (e.g. Amiridis et al., 2009; Giannakaki et al., 2010). Veselovskii et al. (2015) reported that backscatter-related Ångström exponent between 355 and 532 nm is more sensitive to the refractive index, compare to the one between 532 and 1064 nm. In the study of Asian dust, Hofer et al. (2020) showed a larger range of values (-0.5 to 1.8) for the 355-532 nm backscatter-related Ångström exponent compared to the 532-1064 nm one (0.1 to 1.4).

Depolarization ratio at 355 nm can be also estimated, as pollen backscatter at both 355 and 532 nm should be the same under the assumption that the backscatter-related Ångström exponent between 355 and 532 nm for pure pollen is 0. Pollen backscatter contribution at 355 nm ($\chi_{pollen}(355)$) was calculated using lidar-derived particle backscatter coefficient at 355 nm. The inverse model was applied here for the backscatter-related Ångström exponent between 355 and 532 nm (Å(532,355)) and pollen backscatter contribution at 355 nm, using a third parameter $\frac{1}{\eta}$ (as in Eq.6, a function of Å(532,355)), which is proportional to the pollen backscatter contribution at 355 nm, considering the Eq.5 using the wavelength pair of $\lambda_1$=532

and $\lambda_2=355$. Here Å is the backscatter-related Ångström exponent between 355 and 532 nm, for the total particle backscatter coefficient. Under different values of assumed pollen depolarization ratio at 355 nm ($\delta_{x,355}$) from 0.1 to 0.4, linear correlations were found for $\frac{1}{\eta}$ and $\chi_{pollen}(\delta_{x,355}, 355)$ (Fig. S8 in the supplement). Values for $\frac{1}{\eta_{pure}}(\delta_{x,355})$ for 100 % pollen backscatter contribution at 355 nm are reported in Fig. 13, against related $\delta_{x,355}$. Finally, the pollen depolarization ratios at 355 nm of 0.17

and 0.30 were found for IPP-1 (birch) and IPP-3 (pine), respectively (Table 5). Cao et al. (2010) found smaller values with a linear depolarization ratio at 355 nm for paper birch of 0.08, and for Virginia pine of 0.20.

The particle linear depolarization ratio at 355 nm can be calculated by using the pollen depolarization ratio at 355 nm. Mean values of depolarization ratio of pollen layers for IPP-1 and IPP-3 were retrieved and shown in Fig. 14. For both periods, PDR

at 355 nm values are relatively smaller than the ones at 532 nm.

Uncertainty values for pollen depolarization ratios and particle linear depolarization ratio at 355 nm are not given in this paper, as these estimations were under the assumption that the backscatter-related Ångström exponent between 355 and 532 nm for pure pollen is 0, and base on previously retrieved pollen depolarization ratios at 532 nm. More uncertainty sources should be considered for the uncertainty study, and it is complicated to give qualitative values. Nevertheless, a wavelength dependence

seems to be found for depolarization values when pollen is present, which may be a key parameter for pollen recognition and characterization. Thus, depolarization ratio at different wavelengths are needed to identify different pollen types.

## 5    Summary and conclusions

We have defined lidar-derived properties for pure pollen based on a four months pollen campaign, which was performed during May to August 2016 in Kuopio station in Eastern Finland. This station is part of the European Aerosol Research Lidar Network

(EARLINET). Twenty types of pollen were observed and identified by Burkard sampler; among which, birch (*Betula*), pine (*Pinus*), spruce (*Picea*) and nettle (*Urtica*) pollen are most abundant, contributing more than 90 % of total pollen load, regarding number concentrations. Four intense pollination periods (IPPs) were defined considering the pollen seasons and the daily mean pollen concentration values.

Mean values of lidar-derived optical properties in the pollen layer were used to characterise pollen for each IPP. We found

that lidar ratio (LR) values range from 55 to 70 sr for all pollen types, indicating that pollen is medium to high absorbing particles. No significant wavelength-dependence could be determined on LR values using LR at 355 nm and 532 nm, regarding the uncertainties. The wide range of LRs suggest that the LR alone is not a suitable parameter to discriminate between different pollen types. Nonetheless, we showed that the depolarization ratio is the most proper indicator for pollen and further the pollen type, as the depolarization ratio was enhanced when there were pollen in the air, and even higher depolarization ratio was

observed with presence of the more non-spherical spruce and pine pollen. The Ångström exponent could be used to classify different pollen types only under same or similar background conditions, as its value depends a lot on the background aerosols.

As the main results, we provide a novel method for the characterization of pure pollen particles. We present an algorithm to estimate the depolarization values for pure pollen, under the assumption that backscatter-related Ångström exponent between 355 and 532 nm should be zero for pure pollen, as pollen grain are quite large and can be assumed to be wavelength independent at these 2 wavelengths. This algorithm was first tested and validated through a simulator of synthetic lidar profiles

(including a direct model and an inverse model modules). Mathematically, the depolarization ratio for pure pollen can be calculated using the equations given in Sect. 3, if other variables are known or can be assumed. We have developed a retrieval method to estimate the pollen depolarization ratio, which was applied to the lidar observations. The depolarization ratio at 532 nm of pure pollen particles was assessed, resulting to $0.24 \pm 0.01$ and $0.36 \pm 0.01$ for birch and pine pollen, respectively. The uncertainty on assumed backscatter-related Ångström exponent of pure pollen will introduce non-negligible bias in addition

as discussed in Sect. 4.3.1. Pollen optical properties at 1064 nm and 355 nm were also estimated base on retrieved pollen depolarization ratio at 532 nm. The pollen depolarization ratio at 355 nm of 0.17 and 0.30 were found for birch and pine pollen, respectively. The depolarization values show a wavelength dependence for pollen. This can be the key parameter for pollen detection and characterization. Also, a wavelength dependence on the backscatter between 532 and 1064 nm was found, with the value of the backscatter-related Ångström exponent between 532 and 1064 nm of ~0.8 (~0.5) for pure birch (pine) pollen.

Based on simulations in this study, we found that depolarization ratios at 355 nm and 1064 nm would provide valuable information for pollen study, thus more multi-wavelength lidar studies with depolarization characterization on atmospheric pollen are necessary. The presented novel algorithm and the estimated optical properties for pure pollen in this study, provide a good method for pollen characterization and classification. Currently, CALIPSO (Cloud-Aerosol LIdar with Orthogonal Polarization) aerosol type classification scheme includes seven tropospheric aerosol types (Kim et al., 2018, https://www-

calipso.larc.nasa.gov/resources/calipso_users_guide/data_summaries/vfm/index_v420.php, last access: 7 April 2020), in which pollen (or biogenic aerosols in general) is excluded. Such ground-based lidar measurements also provide the possibility to implement a new aerosol type to the CALIPSO classification scheme, for example using the depolarization ratio at 532 nm. This method can also be applied to other aerosol mixtures (e.g., dust and non-dust aerosols) to retrieve the particle linear depolarization ratio related to aerosol types, under the condition that the depolarization ratio of one aerosol type is the only

unknown parameter, and other parameters are known or can be reasonably well approximated. Note that the two constrains mentioned in Sect.3.1 should be considered: both the depolarization ratio and the backscatter-related Ångström exponent of the two aerosol types should be different.

*Data availability*. Lidar data are available upon request from the authors and data "quicklooks" are available on the PollyNET

website (http://polly.tropos.de/, last access: 25 June 2020).

*Author Contributions*. XS analysed the data, developed the algorithm and the simulator, and wrote the paper. XS, EG, MK, and SR conceptualized and finalized the methodology. XS and SB performed the lidar data analysis. AS analyzed the pollen

samples. MK and EG initiated and managed the project. MF, AR, AL, and MK participated in the measurement campaign. All authors were involved in the paper editing, interpretation of the results and discussion of the manuscript.

*Conflicts of Interest*. The authors declare no conflict of interest.

*Special issue statement*. This article is part of the special issue "EARLINET aerosol profiling: contributions to atmospheric and climate research". It is not associated with a conference.

*Acknowledgments.* The authors would like to thank the use of the microphotographs of pollen grains, from PalDat (PalDat – a
10 palynological database, 2000 onwards, www.paldat.org, last access: 7 April 2020), courtesy of the Division of Structural and Functional Botany, University of Vienna. Elina Giannakaki acknowledge the support of Hellenic Foundation for Research and Innovation (H.F.R.I.) under the "First Call for H.F.R.I. Research Projects to support Faculty members and Researchers and the procurement of high-cost research equipment grant (Project number: 2544).

*Financial support.* This work was supported by the Academy of Finland (projects no. 310312 and 329216).

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

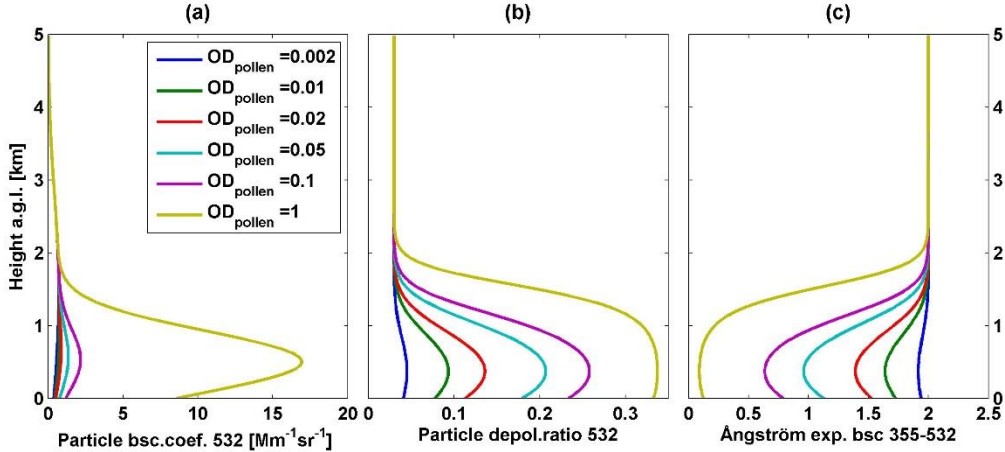

**Figure 1. Six cases of simulated vertical profiles of (a) particle backscatter coefficient at 532 nm, (b) particle linear depolarization ratio at 532 nm, and (c) backscatter-related Ångström exponent between 355 and 532 nm. Simulated results under different input pollen optical depth (OD) values are shown by colour.**

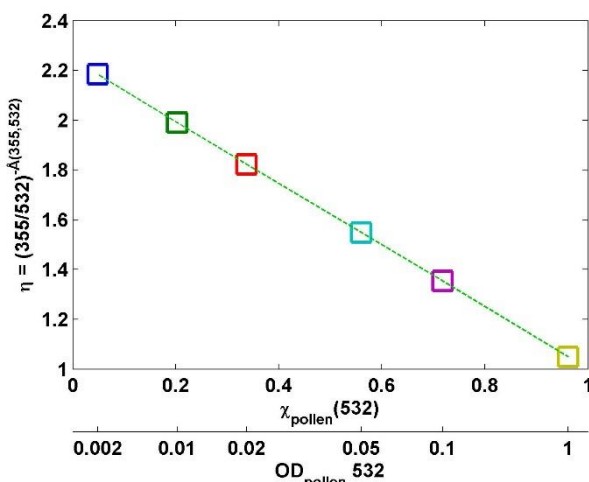

**Figure 2. Scatter plot using the parameter $\eta=(355/532)^{-\text{Å}(355,532)}$ and pollen backscatter contribution ($\chi_{\text{pollen}}$) at 532 nm for 6 simulated cases, of which the input values of pollen optical depth ($OD_{\text{pollen}}$) at 532 nm are defined as 0.002, 0.01, 0.02, 0.05, 0.1, and 1 (shown as the bottom x-axis), and input value of background optical depth is fixed to be 0.1. Mean values of pollen layers (0-1 km) are used for $\chi_{\text{pollen}}$ and $\eta$. They line up perfectly following Eq.5.**

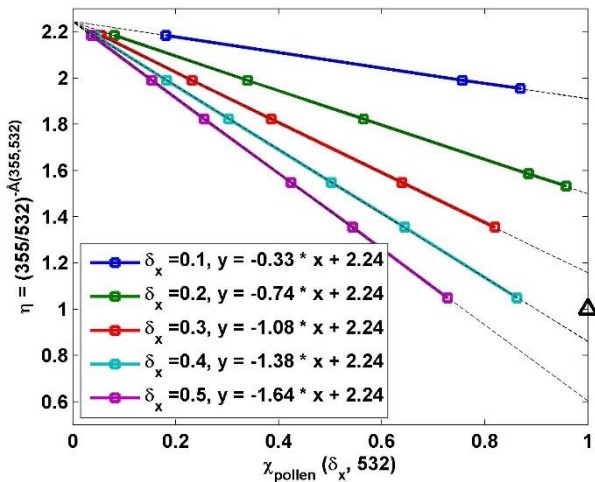

Figure 3. Scatter plots of mean values of η and $\chi_{\text{pollen}}(\delta_x, 532)$ in pollen layer under 5 assumed $\delta_x$ values cases. η is a parameter using backscatter-related Ångström exponent between 355 and 532 nm (Eq.6), and $\chi_{\text{pollen}}(\delta_x, 532)$ is the pollen backscatter contribution at 532 nm inside the pollen layer under a certain assumed pollen depolarization ratio value ($\delta_x$ is 0.1, 0.2, 0.3, 0.4, or 0.5). Linear regression lines are drawn by black dotted lines with fitting equation shown (Eq.5 or 8). The black triangle shows the ideal value: when $\chi_{\text{pollen}}$ is 1, η should be 1 (Å=0).

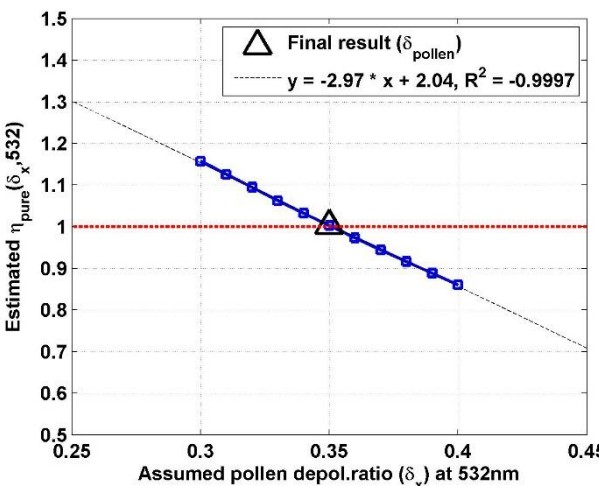

Figure 4. Estimated parameter $\eta_{\text{pure}}$ against the related assumed pollen depolarization ratio $\delta_x$ at 532 nm. $\eta_{\text{pure}}$ is the $\eta(\chi_{\text{pollen}})$ value for the pure pollen (100 % pollen in the observed aerosol particle population, $\chi_{\text{pollen}} = 1$), where η is a parameter using backscatter-related Ångström exponent between 355 and 532 nm (Eq.6). Linear regression line is drawn by black dotted line, with fitting equation shown. The correlation coefficient ($R^2$) value is also given. The final result of 0.35 for pure pollen is found, resulting in $\eta_{\text{pure}}=1$ (i.e. $Å_{\text{pollen}}=0$) (by the black triangle).

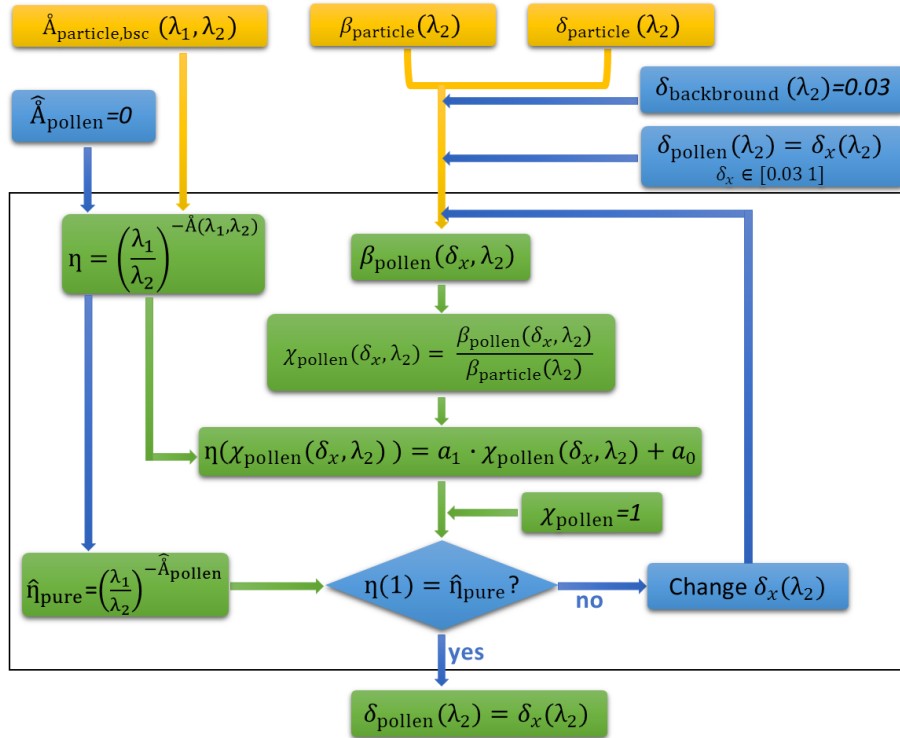

**Figure 5. Flow chart of the inverse model for the retrieval of depolarization ratio value for pure pollen. The orange boxes are for the measured parameters (or simulated output from the direct model), blue boxes for the assumptions/manual input and the green boxes for the estimations/calculations. Detail description is in Sect. 3.2. The wavelength pair $(\lambda_1, \lambda_2)$ is selected as (355,532), (532,355), or (1064,532) in this study.**

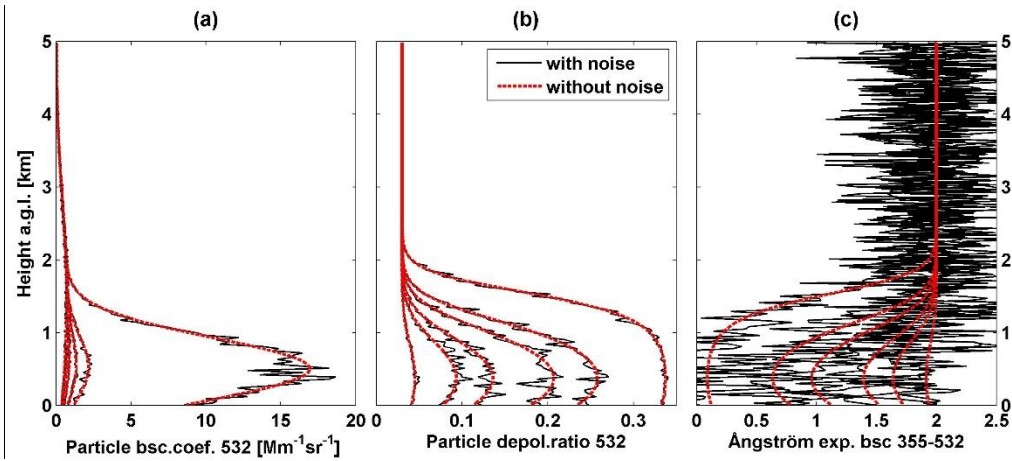

**Figure 6. Example of one group of six simulated profiles of (a) particle backscatter coefficient at 532 nm, (b) particle linear depolarization ratio at 532 nm, and (c) backscatter-related Ångström exponent between 355 and 532 nm. Profiles without noise are shown in red dashed lines, and ones with noise are shown in black lines. Noise levels on backscatter at both 355 and 532 nm were settled as 10 %. Simulated results under 6 input pollen optical depth (OD) values of 0.002, 0.01, 0.02, 0.05, 0.1, and 1 (same as Fig. 1).**

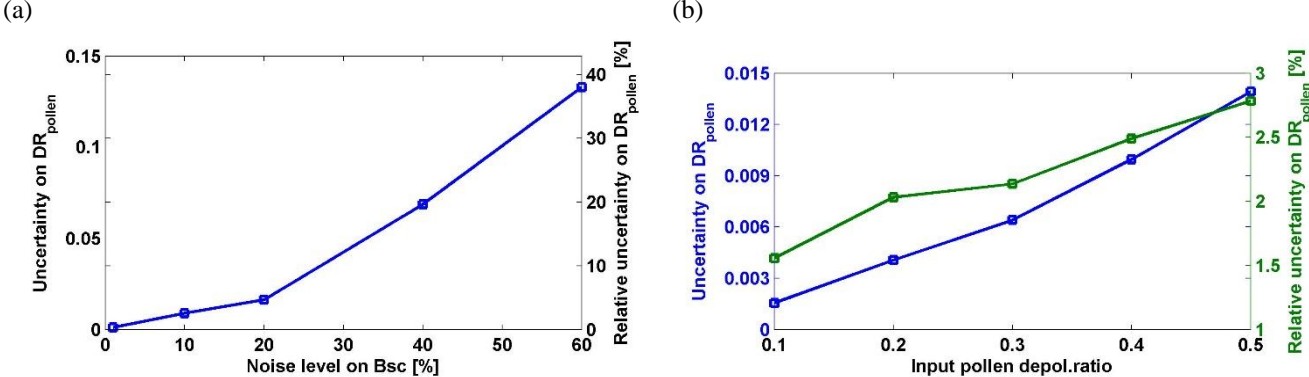

**Figure 7. Examples of estimated uncertainties (left y-axis) and relative uncertainties (right y-axis) on retrieved pollen depolarization ratio (DR_pollen) at 532 nm against (a) the applied noise levels on backscatter coefficient (Bsc), and (b) the initial input values of DR_pollen, using Monte Carlo method. The initial input value of DR_pollen is 0.35 for the example in (a). The noise level on backscatter coefficient (Bsc) is 10% for the example in (b).**

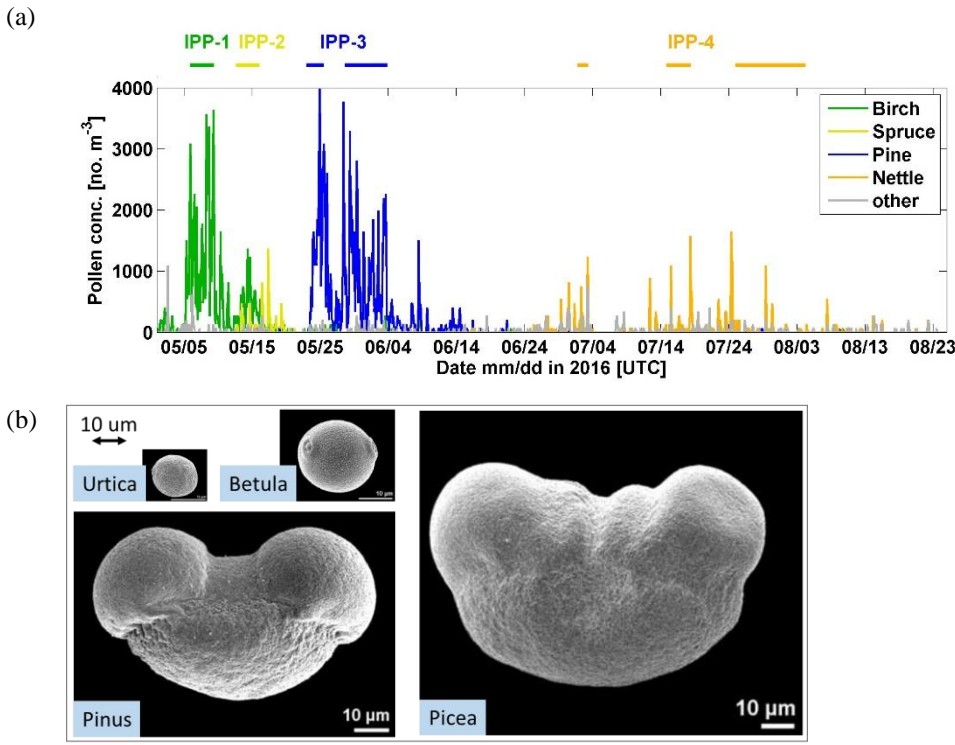

**Figure 8. (a) Pollen concentration (2-hour average) measured by the Burkard sampler at roof level. The main pollen types are shown by colours. Defined intense pollination periods (IPPs) are shown by lines on the top. (b) Microphotographs of pollen grain: *Urtica* (nettle pollen), *Betula pendula* (birch pollen), *Pinus* (pine pollen), *Picea abies* (spruce pollen). Source: PalDat – a palynological database (www.paldat.org, last access: 7 April 2020).**

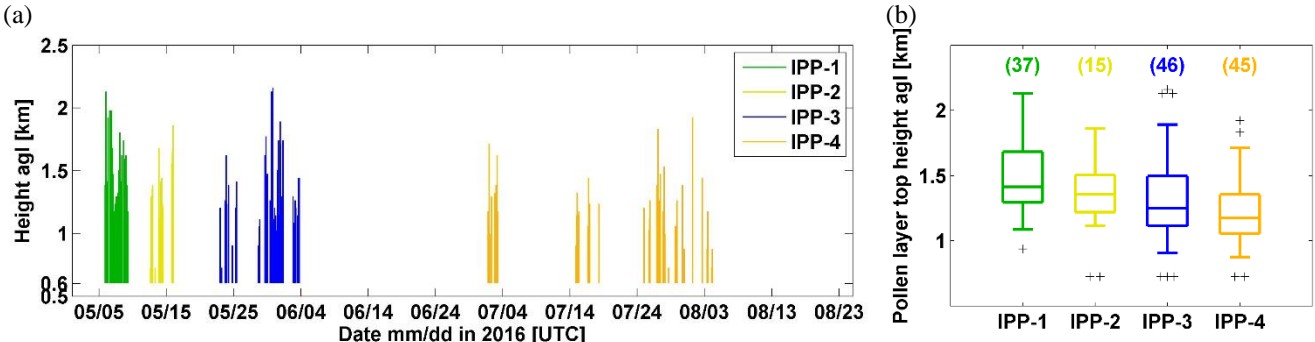

**Figure 9. (a) Pollen layer definition for four intense pollination periods (IPPs). (b) Boxplot of pollen layer top heights during each IPP. Number of available profiles are given. Colours are related to the IPPs.**

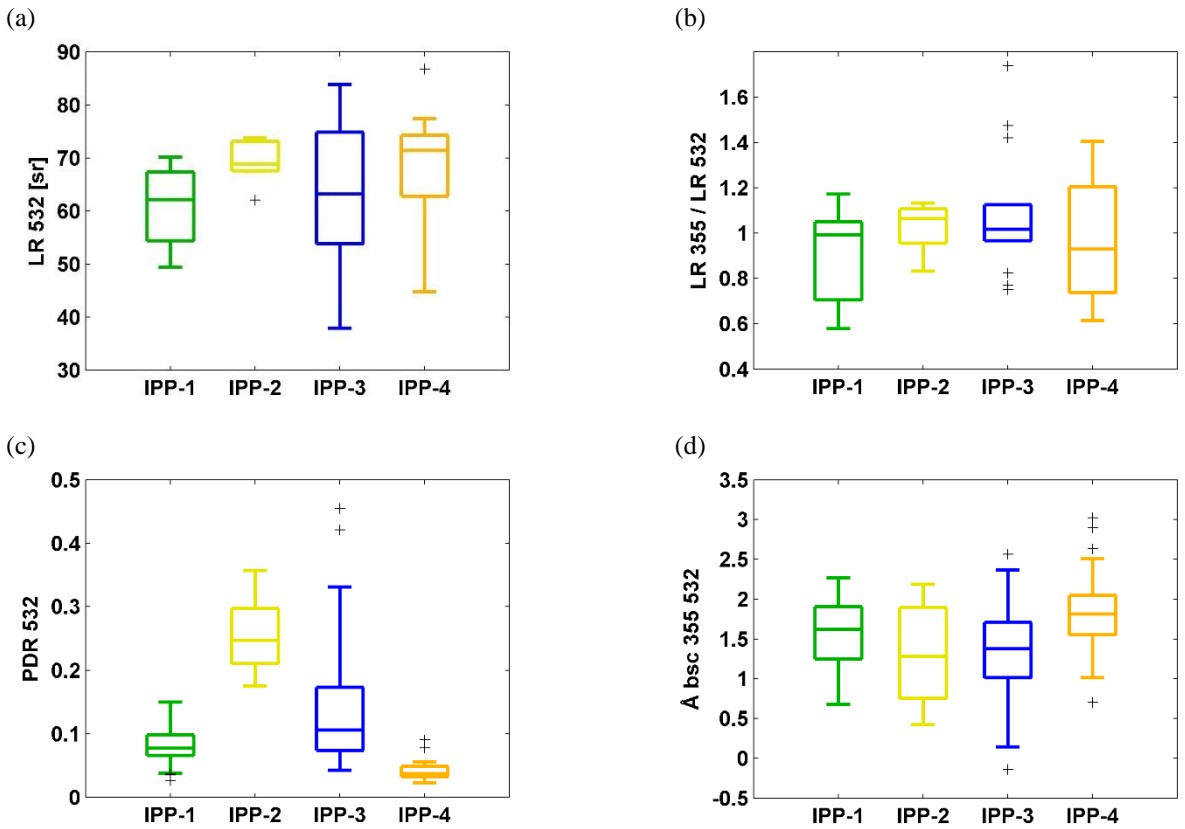

**Figure 10. Boxplots of (a) lidar ratio (LR) at 532 nm, (b) ratio of LR at 355 nm and LR at 532 nm during night-time measurements. Boxplots of (c) particle linear depolarization ratio (PDR), and (c) backscatter-related Ångström exponent between 355 and 532 nm during all-day measurements. Mean values of the detected pollen layer for four IPPs are used. The horizontal line represents the median, the boxes the 25 and 75 % percentiles, the whiskers the standard deviation and the plus signs the outliers.**

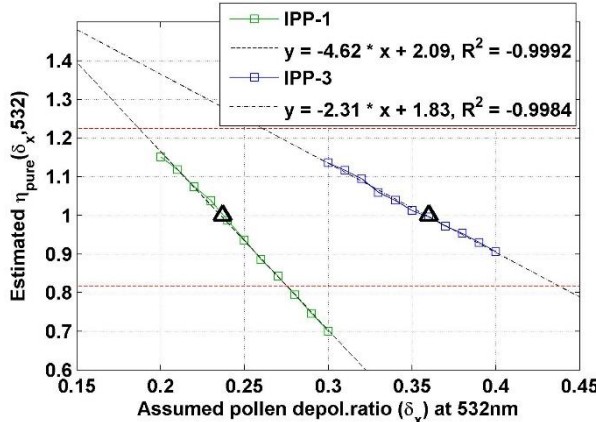

**Figure 11. Estimated $\eta_{pure}$ against the related assumed pollen depolarization ratio $\delta_x$ at 532 nm for IPP-1 (in green) and IPP-3 (in blue). Linear regression lines are drawn by dotted lines, with fitting equations shown. The correlation coefficient ($R^2$) values are also given. $\eta$ is a parameter using backscatter-related Ångström exponent between 355 and 532 nm (Eq.6), and $\eta_{pure}$ is the estimated $\eta$ value for $\chi_{pollen}(\delta_x)=1$ (i.e. pollen contribution in the observed aerosol particle population is 100%) (Eq.8). The final results for pure pollen are shown by the black triangles. $\eta_{pure}$ values of 0.82 and 1.22 (i.e. backscatter-related Ångström exponent of -0.5 and 0.5) are shown by horizontal red dotted lines.**

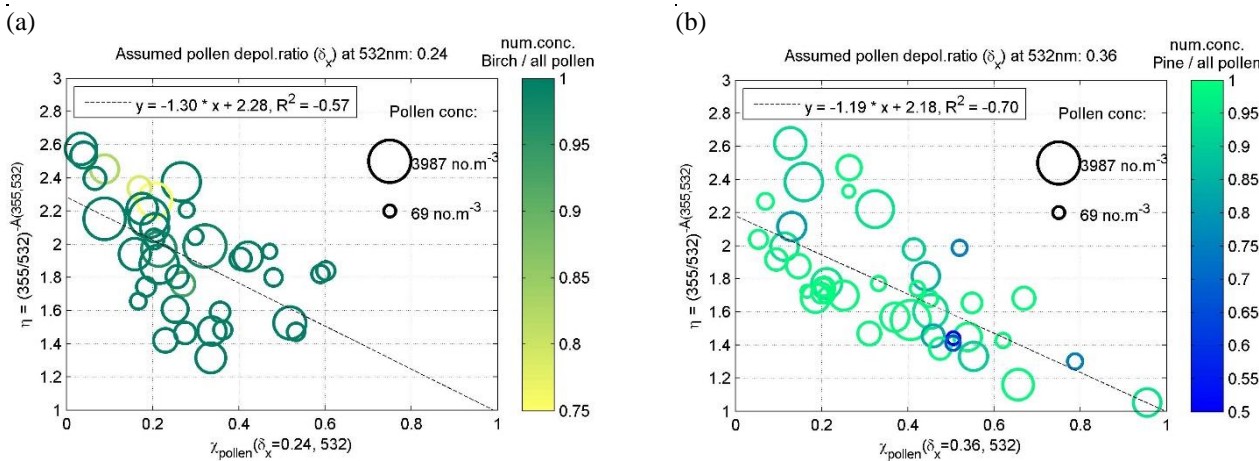

**Figure 12. Mean values of the parameter $\eta$ against pollen backscatter contribution at 532 nm ($\chi_{pollen}(\delta_x, 532)$) inside the pollen layers, during the IPP-1 (a) and IPP-3 (b). $\eta$ is a parameter using backscatter-related Ångström exponent between 355 and 532 nm (Eq.6). The pollen depolarization ratio $\delta_x$ at 532 nm is assumed to be 0.24 for (a) or 0.36 for (b). Linear regression lines are drawn by dotted lines, with fitting equation shown (Eq.5 or 8). The correlation coefficient ($R^2$) is also given. The size denotes the total pollen concentrations measured by the Burkard sampler on roof level; the colour represents the number concentration of the dominant pollen (a: birch, b: pine) against the total pollen number concentration. Similar figures using different assumed values of pollen depolarization ratio can be found in Fig. S5 and Fig. S6 in the supplement.**

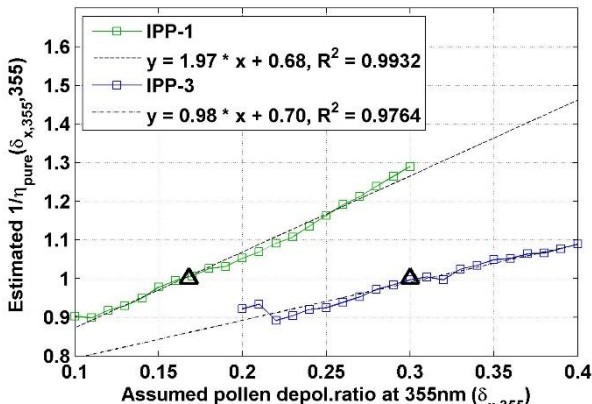

**Figure 13. Estimated $\frac{1}{\eta_{pure}}$ ($\delta_{x,355}$) against the related assumed pollen depolarization ratio at 355 nm ($\delta_{x,355}$) for IPP-1 (in green) and IPP-3 (in blue). Linear regression lines are drawn by dotted lines, with fitting equations shown. The correlation coefficient ($R^2$) values are also given. $\frac{1}{\eta}$ is a parameter using backscatter-related Ångström exponent between 355 and 532 nm (Eq.6), and $\frac{1}{\eta_{pure}}$ is**

5  **the estimated $\frac{1}{\eta}$ value for $\chi_{pollen}(\delta_{x,355}, 355)$=1. The final results for pure pollen are shown by the black triangles. Results are under the assumption that the backscatter-related Ångström exponent between 355 and 532 nm for pure pollen is 0.**

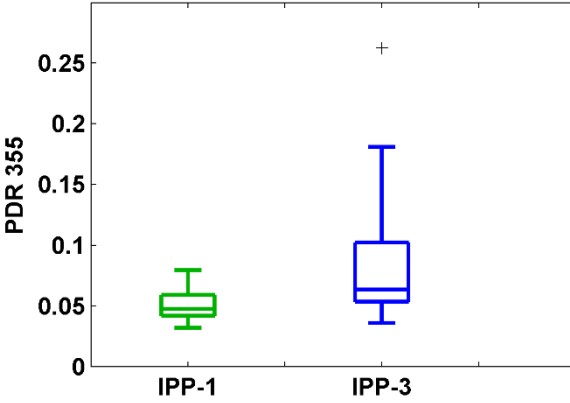

**Figure 14. Boxplots of estimated particle linear depolarization ratio (PDR) at 355 nm. Mean values of the detected pollen layer for**
10  **every IPPs are used. The horizontal line represent the median, the boxes the 25 and 75 % percentiles, the whiskers the standard deviation and the plus signs the outliers. Results are under the assumption that the backscatter-related Ångström exponent between 355 and 532 nm for pure pollen is 0.**

**Table 1. Parameters of pollen and background aerosol layers as input of the direct model. LR: lidar ratio, DR: depolarization ratio, Å bsc: backscatter-related Ångström exponent. A Gaussian distribution is applied for each layer with layer center and half width given.**

| Aerosol type | LR 355nm [sr] | LR 532nm [sr] | DR ($\delta$) 355nm | DR ($\delta$) 532nm | Å bsc 355-532nm | Layer center | half width (Gauss) |
|---|---|---|---|---|---|---|---|
| Pollen | 65 | 65 | 0.35 | 0.35 | 0 | 0.5 km | 1 km |
| Background | 50 | 50 | 0.03 | 0.03 | 2 | 1.5 km | 3 km |

5    **Table 2. The pairs of the parameter η($Å_{particle}$) and $\chi_{pollen}$ at different wavelengths resulting linear relationships are reported.**

| Wavelength pair$(\lambda_1, \lambda_2)$ [nm] | Pollen backscatter contribution at $\lambda_2$ | Backscatter-related Ångström exponent $Å(\lambda_1, \lambda_2)$ | Parameter of $Å(\lambda_1, \lambda_2)$, linearly correlating with $\chi_{pollen}$ | Formulate |
|---|---|---|---|---|
| $\lambda_1=355$ $\lambda_2=532$ | $\chi_{pollen}(532)$ | $Å_{particle}(355,532)$ | $\eta$ | $\eta = \left(\dfrac{355}{532}\right)^{-Å_{particle}(355,532)}$ |
| $\lambda_1=532$ $\lambda_2=355$ | $\chi_{pollen}(355)$ | $Å_{particle}(532,355)$ | $\dfrac{1}{\eta}$ | $\dfrac{1}{\eta} = \left(\dfrac{532}{355}\right)^{-Å_{particle}(355,532)}$ |
| $\lambda_1=1064$ $\lambda_2=532$ | $\chi_{pollen}(532)$ | $Å_{particle}(1064,532)$ | $\eta'$ | $\eta' = \left(\dfrac{1064}{532}\right)^{-Å_{particle}(532,1064)}$ |

**Table 3. (a) Dominant pollen types with their pollen season period, Latin name (Taxa), and typical size. (b) Selected intense pollination periods (IPPs) and the presented dominant pollen types during each IPP. See more descriptions in Sect. 4.1.**

**(a) *Dominant pollen types***

| Pollen type | Pollen season in 2016 (mm.dd – mm.dd) | Taxa | The longest axis size (µm)* |
|---|---|---|---|
| Birch | 04.29-05.26 | *Betula* | 22 - 28 |
| Spruce | 05.13-06.14 | *Picea* | 90 - 110 |
| Pine | 05.23-06.13 | *Pinus* | 65 - 80 |
| Nettle | 06.27-08.14 | *Urtica* | 15 - 20 |

**(b) *Selected intense pollination periods (IPPs)***

| IPP | Period time in 2016 (mm.dd – mm.dd) | Pollen types (percentage of number concentration) |
|---|---|---|
| IPP-1 | 05.05-05.09 | Birch (97%), other pollen (3%) |
| IPP-2 | 05.12-05.16 | Birch (82%), Spruce (14%), other pollen (4%) |
| IPP-3 | 05.23-05.25 & 05.28-06.03 | Pine (95%), other pollen (5%) |
| IPP-4 | 07.01-07.03 & 07.14-07.18 & 07.24-08.04 | Nettle (75%), other pollen (25%) |

* Values from Nilsson et al., 1977.

**Table 4. Lidar derived optical values of pollen layer for the intense pollination periods (IPPs) (mean values ± standard derivation are given). LR: lidar ratio, PDR: particle linear depolarization ratio, Å bsc: backscatter-related Ångström exponent.**

|        | Raman cases | LR 355nm [sr] | LR 532nm [sr] | All cases | PDR 532nm | Å bsc 355nm-532nm |
|--------|-------------|---------------|---------------|-----------|-----------|-------------------|
| IPP-1  | 10          | 54 ± 12       | 61 ± 8        | 37        | 0.08 ± 0.03 | 1.57 ± 0.43     |
| IPP-2  | 7           | 71 ± 10       | 69 ± 4        | 15        | 0.25 ± 0.06 | 1.32 ± 0.61     |
| IPP-3  | 13          | 66 ± 12       | 63 ± 14       | 46        | 0.14 ± 0.09 | 1.38 ± 0.57     |
| IPP-4  | 15          | 63 ± 14       | 68 ± 11       | 45        | 0.04 ± 0.01 | 1.83 ± 0.43     |

**Table 5. Linear depolarization ratios for pure pollen. The assumption of backscatter-related Ångström exponent between 355 and 532 nm for pollen ($Å_{pollen}$) should be 0 was applied for this study. The uncertainty on $Å_{pollen}$ was not taken into account for the standard deviation shown here, which may introduce non-negligible additional bias. For example, under the assumption of $Å_{pollen} ± 0.5$, the range of depolarization ratio values is 0.19 to 0.27 for birch pollen and 0.26 to 0.44 for pine pollen. See more details in Sect. 4.3.**

|                                           | Pollen type   | Depolarization ratio at 532 nm | Depolarization ratio at 355 nm |
|-------------------------------------------|---------------|--------------------------------|--------------------------------|
| This study, Finland (in the atmosphere)   | Silver birch  | 0.24 ± 0.01                    | 0.17                           |
|                                           | Scots pine    | 0.36 ± 0.01                    | 0.30                           |
| Cao et al. (2010), Canada (in an aerosol chamber) | Paper birch | 0.33 ± 0.004             | 0.08 ± 0.008                   |
|                                           | Virginia pine | 0.41 ± 0.006                   | 0.20 ± 0.013                   |