# Peer review of "Optical characterization of pure pollen types using a multiwavelength Raman polarization lidar"

_Atmospheric Chemistry and Physics, 2020_

## Referee Comment (RC1) · Anonymous Referee #1 · 21 Aug 2020

The authors describe a methodology to infer the particle linear depolarisation ratio (PLDR) of different types of airborne pollen in their pure state, i.e. unmixed with ambient aerosols. Due to the size and irregular shape of pollen, it can be expected that their PLDR is larger than that of most aerosol types though likely smaller that that of mineral dust. Knowing the PLDR of undiluted aerosol types is important as it allows for separating, in the presented cases, the contribution of pollen to pollen-containing aerosol mixtures.

I am not quite sure that the authors are actually presenting simulations or modelling in their discussion of a simulator. It all seems quite analytical. To my understanding,

the shape of the extinction coefficient profiles for pollen and the background aerosol is defined as given in Table 3. The magnitude of the extinction coefficient of both species is then determined by the respective optical thickness which is also set. Backscatter coefficients are obtained using the set lidar ratio and the Angstrom exponent is derived trough mixing the set values of the two aerosol types. It is totally unclear, though, how the authors arrive at the mixed PLDR profiles presented in Figs. 4 and S2b. PLDR values for both types are also set in Table 3 but the mixing rule is not linear such as summing up the backscatter or extinction coefficients. Is this when simulations come into play? If so, what is done to get the PLDR profiles? It might be that the authors have used Eq. (6) which is given in Section 3.3.2 to obtain PLDR profiles of the pollen-background-aerosol mixture. If so, the entire simulator would be circular as identical calculations would be done in both directions. This would explain the perfectly linear relationships presented in Figs. 5, 6, and 7. I am afraid I cannot assess the scientific quality of this work before the authors clarify the description of the simulator, particularly with respect to the points made above.

**Further comments**

- The title is a bit misleading as the reader might expect observations from an aircraft. I'd suggest a clearer title such as "Optical characterisation of airborne pollen from lidar measurements in Finland"

- The part about CALIPSO in the Introduction (page 2, lines 21-28) should be omitted. It is not needed as there is no later reference on how to apply the new results of this work to improve the CALIPSO aerosol typing.

- page 3, lines 3-13 are more suitable in the introduction

- I'd suggest a change of the structure of the paper: 1. Introduction, 2. Site and instruments, 3. Lidar simulator, 4. Results. Such a structure allows for a clear

separation of instruments, methods, and findings. In addition, the lidar parameters should already be introduced in the description of the lidar. This way, the reader knows what's available for the theoretical studies in the next section.

- Are there any objective criteria for determining the exact times of the different intensive pollination periods? I am thinking of a certain threshold of the extinction coefficient at a certain height or similar quantifiable criteria.

- It would be good to get some information on how often pollen are observed at such high altitudes and how they get up there in the first place.

- The abbreviation PBC is not ideal as it is often used to denote the particle backscatter coefficient. I'd suggest to rename this into some ratio with a different variable.

- There is no mentioning of the assumed shape of the pollen and background aerosol profiles in the uncertainty study. Is there any justification for selecting this shape? How general are they? Is the same pollen profile assumed in the inverse model?

- Why is the pollen layer set to extend from 0 to 1 km when the measurements (1) only start at 600 m or so and (2) show pollen all the way to 2 km height?

- It would also be good to get an idea of typical values of total and pollen-related optical thickness at your site to assess the choice of values in your method. Is a pollen optical depth of unity even possible?

---

## Referee Comment (RC2) · Anonymous Referee #2 · 27 Aug 2020

This paper studied the optical properties of pollen using multi-wavelength Raman Lidar. Like dust particles, it was the first to suggest a method of classifying pollen from atmospheric aerosols and calculating the optical depth, lidar ratio, and depolarization polarization of pure pollen only. This method can be applied only under conditions where there are no dust particles in the atmosphere, but it is considered to be a very important study because it is a method that can calculate information on the distribution and concentration of pollen with a spatial distribution using remote sensing technology. It is judged that the thesis is well structured and explained in detail the new method and process. It is considered acceptable to publish the paper as it is. However, as a suggestion, in this paper, the study results were calculated by applying the method

proposed in this study only for the two observation periods (IPP-1 and IPP-3) among four periods of Birch and Pine pollen. How about showing the results by applying this method for the period of IPP-2 and IPP-4? In this case, not only Birth and Pine, but also other types of pollen or a mixture of various types, couldn't we derive meaningful research results?

---

## Author Comment (AC1) · 4 Sep 2020

**Response to Referee #1**

Thank you for carefully reading the manuscript and providing useful suggestions to improve the paper. The replies to the referee comments are given below. The referee comments are highlighted in blue with our responses in black. The sentences in the manuscript are in Italic, with the modifications in the revised manuscript in red.

**The authors describe a methodology to infer the particle linear depolarisation ratio (PLDR) of different types of airborne pollen in their pure state, i.e. unmixed with ambient aerosols. Due to the size and irregular shape of pollen, it can be expected that their PLDR is larger than that of most aerosol types though likely smaller that that of mineral dust. Knowing the PLDR of undiluted aerosol types is important as it allows for separating, in the presented cases, the contribution of pollen to pollen-containing aerosol mixtures.**
**I am not quite sure that the authors are actually presenting simulations or modelling in their discussion of a simulator. It all seems quite analytical.**
We are presenting a simulation in this paper. In the simulator section, we applied the algorithm to the synthetic lidar data. This part will be clarified in the revised version.

In the revised manuscript, Page 6 lines 4-6 will be:

*This algorithm is first tested through a simulator (Sect. 3.3.2) using the synthetic lidar data, and then applied to the real lidar observations (Sect. 3.4). The simulator includes a direct model and an inverse model modules (the block diagram is shown in Fig. S1 in the supplement); Similar ones have already been used for forest and aerosol studies (Shang et al., 2018; Shang and Chazette, 2015). Synthetic data are used in this section to present our methodology.*

**To my understanding, the shape of the extinction coefficient profiles for pollen and the background aerosol is defined as given in Table 3. The magnitude of the extinction coefficient of both species is then determined by the respective optical thickness which is also set. Backscatter coefficients are obtained using the set lidar ratio and the Angstrom exponent is derived trough mixing the set values of the two aerosol types.**
Yes, this is the way we simulate the extinction and backscattering coefficients.

**It is totally unclear, though, how the authors arrive at the mixed PLDR profiles presented in Figs. 4 and S2b. PLDR values for both types are also set in Table 3 but the mixing rule is not linear such as summing up the backscatter or extinction coefficients. Is this when simulations come into play? If so, what is done to get the PLDR profiles? It might be that the authors have used Eq. (6) which is given in Section 3.3.2 to obtain PLDR profiles of the pollen-background-aerosol mixture. If so, the entire simulator would be circular as identical calculations would be done in both directions. This would explain the perfectly linear relationships presented in Figs. 5, 6, and 7. I am afraid I cannot assess the scientific quality of this work before the authors clarify the description of the simulator, particularly with respect to the points made above.**
We will clarify the PDR calculation in the revised version. The calculation is as following:

We follow the detailed calculations in Tesche et al. 2009.
The particle depolarization ratio ($\delta_{particle}$) is expressed as Equation 4 in the manuscript:
$$\delta_{particle} = \frac{\beta_{pollen}^{\perp} + \beta_{background}^{\perp}}{\beta_{pollen}^{\parallel} + \beta_{background}^{\parallel}}, \tag{4}$$
The depolarization ratio of one particle type can be defined as:
$$\delta_x = \frac{\beta_x^{\perp}}{\beta_x^{\parallel}}, \tag{S1}$$

The index *x=pollen* or *background* denotes the contribution of pollen or background particles, respectively. We can use the following relationships mathematically:

$$\beta_x = \beta_x^{\perp} + \beta_x^{/\!/} ,\tag{S2}$$

$$\beta_x^{/\!/} = \frac{\beta_x}{1+\delta_x} ,\tag{S3}$$

$$\beta_x^{\perp} = \frac{\beta_x \delta_x}{1+\delta_x} ,\tag{S4}$$

We replace equations S3 and S4 in equation 4, the particle depolarization ratio can be then calculated using the particle backscatter coefficients ($\beta_{pollen}$ and $\beta_{background}$) and the depolarization ratios of both particle types ($\delta_{pollen}$ and $\delta_{background}$):

$$\delta_{particle} = \frac{\frac{\beta_{pollen}*\delta_{pollen}}{\delta_{pollen}+1}+\frac{\beta_{background}*\delta_{background}}{\delta_{background}+1}}{\frac{\beta_{pollen}}{\delta_{pollen}+1}+\frac{\beta_{background}}{\delta_{background}+1}} ,\tag{S5}$$

These equations and descriptions will be added in the supplement for the revised version.
In the revised manuscript, Page 6 lines 24-26 will be:

*Next, pollen layer and background layer are summed up, and then the vertical profiles of aerosol backscatter coefficient,  lidar ratio and Ångström exponent of the total aerosols are simulated (e.g., Fig. S2b); Vertical profiles of particle depolarization ratio can be also calculated following eq.S5 in the supplement (the detailed calculation is also given). Theoretically, these parameters can be derived directly from lidar observations.*

**Further comments**
**• The title is a bit misleading as the reader might expect observations from an aircraft. I'd suggest a clearer title such as "Optical characterisation of airborne pollen from lidar measurements in Finland"**
Thank you for the suggestion, we will change the title as:

*Optical characterization of pure pollen types using a multi-wavelength Raman polarization lidar.*

We have also removed "in Finland" as the presented method can also be applied to other sites.

**• The part about CALIPSO in the Introduction (page 2, lines 21-28) should be omitted. It is not needed as there is no later reference on how to apply the new results of this work to improve the CALIPSO aerosol typing.**
We agree. We will remove the related information in the introduction for the revised version.

**• page 3, lines 3-13 are more suitable in the introduction**
We agree that these sentences can also be placed in the introduction, but we think these sentences are more suitable for the description of our campaign site. So we decide to keep them in the section 2.

**• I'd suggest a change of the structure of the paper: 1. Introduction, 2. Site and instruments, 3. Lidar simulator, 4. Results. Such a structure allows for a clear separation of instruments, methods, and findings. In addition, the lidar parameters should already be introduced in the description of the lidar. This way, the reader knows what's available for the theoretical studies in the next section.**
Thank you for the suggestion, we were considering the suggested structure, which is also good. Still we decided to keep the current structure as we find it logical for the purpose.

In the section 3 Methodology and results, we present first the results from Burkard sampler (Sect. 3.1), based on which we define the intense pollination periods. Secondly, we present the optical properties of the pollen layer (Sect. 3.2). Which are a quasi-straightforward results and can be easily retrieved from lidar observations. Thirdly, we present the optical properties of pure pollen (Sect. 3.3 and 3.4), first by the synthetic simulation then applied to the lidar measurements. In this third part, we introduce the algorithm along with the assumptions.

The optical properties retrieved from lidar were presented in section 2, page 3 lines 16-24.
We will add more descriptions in section 2 as (page 3 lines 16-24):

*Polly$^{XT}$ has three emission wavelengths (355, 532 and 1064 nm) and seven detection channels (including three emitted wavelengths channels, three inelastic Raman-shifted wavelengths channels (387, 407 and 607 nm) and the cross-polarization channel at 532 nm). During daytime, the Klett-Fernald method (Fernald, 1984; Klett, 1981) is applied using the elastic signals to retrieve the extinction coefficient which describes the combined effect of particle absorption and scattering, and the backscatter coefficient which describes particle backscattering at 180° scattering angle. During night-time, profiles of extinction and backscatter coefficients at 355 and 532 nm can be derived independently using elastic and inelastic Raman-shifted wavelengths (387 and 607 nm), based on the Raman inversion (Ansmann et al., 1992). The ratio of extinction to backscatter coefficient is called lidar ratio (LR), which is considered an important parameter to separate particle types, as it depends on their single scattering albedo and backscatter phase function, thus being a function of size distribution and chemical composition. The cross- and total- polarization channels of the Polly$^{XT}$ allow the retrieval of the volume depolarization ratio (VDR) and linear particle depolarization ratio (PDR) at 532 nm, which provide information on the shape of the scattering particles. Multi-wavelength measurements (355 nm, 532 nm and 1064 nm) enable the determination of Ångström exponents between each wavelength pairs, which are related to the particle nature, mostly the size. The operated lidar system has an initial spatial resolution of 30 m and a temporal resolution of 30 s.*

**• Are there any objective criteria for determining the exact times of the different intensive pollination periods? I am thinking of a certain threshold of the extinction coefficient at a certain height or similar quantifiable criteria.**
In this study, we mainly considered the pollen concentration measured by Burkard, the daily mean pollen concentrations were used as a constraint. We will add this information in the revised version as (page 4 lines 24-25):

*Four intense pollination periods (IPPs) are defined considering  the pollen seasons and the daily mean pollen concentration values of these 4 dominant pollen types (Table 1). A minimum value of 300 no. m$^{-3}$ were applied for birch and pine pollen for IPP-1 and IPP-3, whereas a smaller value of 20 no. m$^{-3}$ were applied for spruce and nettle pollen for IPP-2 and IPP-4. In addition, the availability of  lidar measurements were considered for the IPP definition.*

This method is good enough for our measurement site, and can be used for the other sites under the condition that there is no other depolarization particles. For some sites, more criterions would be needed to define a good pollination period, for example using back-trajectory or in situ measurements to avoid dust effect periods/layers. In this study, we considered the 1$^{st}$ aerosol layer near ground using the layer definition method.

**• It would be good to get some information on how often pollen are observed at such high altitudes and how they get up there in the first place.**
That's a very good point. And it is included in our future work plan. In our other campaigns, we used the drone measurements for such study. But this part is not in the scope of the presented study.

We have explained the long distance transport of pollen in our previous paper (last paragraph of section 4.1 in Bohlmann et al. 2019), and we will add the sentence in the revised version as (Page 2, line 9):

*Several studies on the long distance transport of pollen (Rousseau et al., 2008; Skjøth et al., 2007; Szczepanek et al., 2017) have shown that pollen grains can be lifted up to several kilometers and be dispersed by wind over thousands of kilometers.*

**• The abbreviation PBC is not ideal as it is often used to denote the particle backscatter coefficient. I'd suggest to rename this into some ratio with a different variable.**
Thank you for point this out! We agree, and we will change this PBC to $\chi_{pollen}$ for the whole manuscript (including figures).

For example in Page 6 lines 29-31:

*Pollen backscatter contribution  inside the pollen layer from heights z1 to z2 (in this simulation z1 = 0, z2 = 1 km), denoted as $\chi_{pollen}$, is defined as the ratio of pollen backscatter coefficient ($\beta_{pollen}$) and the total particle backscatter coefficient ($\beta_{particle}$). Note that the use of "particle" here is to distinguish from "molecular".*

$$\chi_{pollen}(z1, z2) = \frac{\int_{z1}^{z2} \beta_{pollen}}{\int_{z1}^{z2} \beta_{particle}}, \tag{1}$$

**• There is no mentioning of the assumed shape of the pollen and background aerosol profiles in the uncertainty study. Is there any justification for selecting this shape? How general are they? Is the same pollen profile assumed in the inverse model?**
There is no assumption of profiles for the inverse model, as it uses the output of direct model as its input.
We used the same shape of profiles as presented in section 3.3.1. In the section 3.3.3 (page 8, Line 25), we mentioned "*using the parameters of previous simulated 6 cases (Sect. 3.3.1)*".

We agree that it was not very clear. We will clarify this in the revised version as (Page 8 line 25):

*The uncertainty study of this method is investigated in this section. The input parameters of the direct model are defined in 3.3.1, with optical depth (OD) of the background aerosol of 0.1, and pollen OD of 0.002, 0.01, 0.02, 0.05, 0.1, or 1. Nonetheless, some input parameters (e.g., the pollen depolarization ratio $\delta_{pollen}$ and the backscatter-related Ångström exponent for pollen $\widehat{A}_{pollen}$) were selected as different values for different uncertainty studies, which are clarified in each paragraph. The output of each direct model simulation were then used as the input of the inverse model.*

Also in Page 9 line 10 will be modified as:

*The parameters for the 6 cases simulated earlier  (as defined in Sect.3.3.1, with values given in Table 3) are used again  in this simulation*

**• Why is the pollen layer set to extend from 0 to 1 km when the measurements (1) only start at 600 m or so and (2) show pollen all the way to 2 km height?'**
This is the simulation of synthetic lidar data, the results don't change if the assumed layer heights change. In the simulation we assumed a pollen concentration with a layer center of 0.5km and half width of 1km. This is why we defined the pollen layer as 0-1 km. Theoretically, we can change the input values of the direct model as whichever reasonable values; in the given example, we just simulate one case. It is also possible to simulate a case with pollen concentration from 600m to ~2km, as the ones

from the real measurement pollen layer. But the conclusion of the simulation section would remain the same.

We will clarify this in the revised version as (Page 6, line 17):

*In addition, the conclusion of the simulation section is not depended on the assumed profile shape or height.*

**• It would also be good to get an idea of typical values of total and pollen-related optical thickness at your site to assess the choice of values in your method. Is a pollen optical depth of unity even possible?**
This is a good point, and we are working on it. We are collecting more data, so as to provide more statistic values.

---

## Referee Comment (RC3) · Anonymous Referee #1 · 8 Sep 2020

Thank you for your quick reply to my Referee comment. I now have a better picture of what you are doing. I also believe that your presentation is unnecessarily convoluted. The manuscript should be simplified in the presentation of the method as well as with regard to the used language and parameters. For instance, it would be much easier to follow your reasoning if you were to use the Ångtröm exponent $\mathring{A}$ in your method rather than the unphysical parameters $\eta$, $\eta'$, $\eta''$, and $\widehat{\eta}$. I had to continuously go back and forth to remind myself what all those parameters represent. I also don't agree with your reference to simulations, a direct model, and an indirect model. What your present is a purely analytical treatment of synthetic and measured lidar profiles.

[Figure]

**Here is a description of your method as I understand it:**

You are defining a set of synthetic lidar profiled for pollen and background aerosol using the set profile shape and the parameters in Table 3. These profiles are then combined to obtain a profile for the mixture of the two. You use Eq. (S5), which is the same as Eq. (13) in *Tesche et al.* (2009), to get the particle linear depolarisation ratio of the mixture. Your Eq. (S5) is transformed to Eq. (6) / Eq. (14) in *Tesche et al.* (2009) by substituting $\beta_{\text{pollen}} = \beta_{\text{total}} - \beta_{\text{background}}$. You now have full knowledge of the system and can calculate the pollen ratio $\chi = \beta_{\text{pollen}}/\beta_{\text{total}}$. Finally, you show that the relationship between $\mathring{A}$ and $\chi$ can also be analytically described to find the value of $\mathring{A}$ related to $\chi = 1$. Per definition in Table 3, $\mathring{A}$ is zero for $\chi = 1$ in your synthetic data.

**Comments:**

- I understand that your choice of parameters in Table 3 is not critical for presenting the overall approach. Nevertheless, it would be nice to get an idea of why those specific values have been selected. In particular, I find the background Ångtröm exponent of 3 quite large.

- Note that I am using the Ångtröm exponent in my description as I find it much easier to follow the steps using a parameter that bears physical meaning.

- You probably don't even need to include the vertical integration of $\chi(z)$. If you use the profile of $\chi(z)$ from your synthetic data, they will still line up perfectly. In the application to real-life measurements, you might also want to leave out the vertical integral as this would require initial knowledge of the pollen layer extend in the measurements. I have a feeling that values of $\chi(z)$ outside the pollen layer will be easy to recognise and screen out in the display of $\mathring{A}$ over $\chi$.

**Next steps in the methodology:**

[Figure]

Now that you know everything about your model aerosol, you basically turn around and use the same set of equations in the other direction with $\delta_{\mathrm{pollen}}$ as the unknown parameter. I would expect an inverse model to be completely independent from the earlier calculations. Instead, your just re-shuffle the equations used before, vary the input value of $\delta_{\mathrm{pollen}}$, and iterate until you have found the value of $\delta_{\mathrm{pollen}}$ for which $\mathring{A} = 0$ at $\chi = 1$. It's as simple as that but it took me quite a while to get there based on your description. I'd therefore encourage you to simplify the presentation of your methodology.

**Now some more comments regarding the manuscript:**

- I again strongly encourage you to separate the presentation of the methodology from the presentation of the results. This is customary in scientific writing and allows the reader not only to better follow your reasoning but also to separate more general relations from your specific results.

- I am quite sceptical about Section 3.4.2. You have defined no profile of $\beta_{1064}$ and no $\mathring{A}_{532/1064}$ in your synthetic data set. How could you know how to interpret your findings when applying this extended approach to real-life data? Your retrieval of $\delta_{\mathrm{pollen}}$ at 355 nm is basically analogous to that at 532 nm. In fact, the choice of values in Table 3 indicates that profiles at 355 and 532 nm should be identical. Why not use the same method at 355 and 532 nm? This should already be discussed in the theory section.

- While the information on pollen type and concentration in Figures 11, S4, S5, and S8 is certainly good to have, it is not needed in those plots. Instead, they distract from the intended message. As stated above, I'd expect that the display would work just the same using all values of $\chi(z)$. The ones outside the pollen layer should be easy to identify as (strong?) deviations from the desired relationship. Using the integrated parameter with the actual measurements might reduce signal noise. However, it requires knowledge of the base and top of the pollen layer as you don't want to include values of $\chi(z)$ outside of this layer in your integration.

- You might want to state that this method can also be applied to other aerosol mixtures to retrieve the particle linear depolarisation ratio related to aerosol types that are dominated by coarse particles ($\mathring{A}_{355/532} = 0$ needs to be fulfilled), as long as the particle linear depolarisation ratio of the second aerosol types is known or can be reasonably well approximated. An obvious application would be the retrieval of the particle linear depolarisation ratio related to undiluted mineral dust from different source regions. The lidar measurements for such a retrieval could be performed further away from the source regions, which translates into a strong reduction of logistical effort.

---

## Referee Comment (RC4) · Anonymous Referee #3 · 29 Sep 2020

The authors present multi-wavelength Raman polarization lidar measurements of pollen layers in Finland combined with a Burkard pollen sampler. Active remote sensing measurements of pollen are rarely found in literature. Therefore, the present manuscript enriches our knowledge about the optical properties of abundant pollen types such as birch and pine pollen. Northern Europe (Finland) is a good location for such a study as it is less affected by other depolarizing aerosol particles such as mineral dust. Additionally, the authors present a novel approach to derive the depolarization ratio of pure pollen layers. Although it is related to some uncertainties, it is a big step forward compared to just presenting the layer mean values. I support the idea that measurements of the depolarization ratio at various wavelengths should be enforced in

future pollen-related studies. Polarization lidars may in future support pollen forecasts and help citizens with pollen allergy thanks to the characterization of pure pollen types by these authors. The quality of the figures and tables is high.

Finally, I recommend publication after minor revisions.

Major remarks:

1. You use a value of 3 for the backscatter-related Ångström exponent of the background aerosol. Do you have any statistical evidence of this value for the station at Kuopio? Is it a mean value for the pollen-free periods? And how sensitive is your analysis to this assumption?

2. Your novel approach for getting the depolarization ratio of the pure aerosol type is remarkable. I am just wondering whether the mixture of continental background aerosol and pollen has a significant effect on the lidar ratio, too. It would be great to have the lidar ratio and the depolarization ratio for pure birch and pine pollen at the end. Please comment on this.

Minor remarks:

3. P5,L25: "The extinction-related and backscatter-related Ångström exponent were also retrieved for pollen layers." – Is the extinction-related Ångström exponent shown somewhere? It must not be shown in the manuscript, some descriptive words are sufficient.

4. P10,L30 The Ångström exponent is related to extinction or backscatter?

5. P11,L10 Are the measurements presented by Cao et al., (2010) performed at exactly $180°$ backscatter direction? This is not so easy to achieve in chamber experiments. Maybe there is an additional source for the discrepancy arising from the optical design of the Cao measurements?

6. Fig. 1+2 and Tab. 1: Please provide the year (2016) whenever you provide dates.

Do it in the caption or just like this "Date mm/dd in 2016 [UTC]".

7. How do you get to the uncertainty range +/-5% for pine pollen? Varying the Ångström exponent by +/- 0.5 leads to values of 26 to 44% (Fig. 12 and P11,L9).

Technical remarks

- Affiliations: "P.O. Box 1627, 5 70211" – seems not necessary and isn't provided for the other institutes

- P1,L11 / P2,L32: depolarization ratio values/value

- P3,L17: volume linear depolarization ratio (VDR) and particle linear depolarization ratio (PDR)

- P4,L18: spoken communication – with whom? Please acknowledge the name of the person

- P6,L10: non-depolarizing aerosol – the received light is depolarized, but the aerosol is depolarizing, please change it throughout the manuscript

- P6,L12+L30: this type of indices should not be written in italic – please change it throughout the manuscript

- P6,L21: "thus six pollen backscattering are simulated." – backscatter coefficients or backscatter coefficient profiles (similar P12,L8)

- P9,L8/9: It would be a good idea to begin a new paragraph with line 9

- Fig. 1, caption of y-axis: [no m-3] – it is -3

- Fig. 3a, caption of y-axis: LR 532 [sr] – unit is missing

---

## Author Comment (AC2) · 25 Oct 2020

**Response to Referee #2**

Thank you for carefully reading the manuscript and providing useful suggestions to improve the paper. The replies to the referee comments are given below. The referee comments are highlighted in blue with our responses in black.

*This paper studied the optical properties of pollen using multi-wavelength Raman Lidar. Like dust particles, it was the first to suggest a method of classifying pollen from atmospheric aerosols and calculating the optical depth, lidar ratio, and depolarization polarization of pure pollen only. This method can be applied only under conditions where there are no dust particles in the atmosphere, but it is considered to be a very important study because it is a method that can calculate information on the distribution and concentration of pollen with a spatial distribution using remote sensing technology. It is judged that the thesis is well structured and explained in detail the new method and process. It is considered acceptable to publish the paper as it is.*
Authors thank the reviewer for the positive comments.

*However, as a suggestion, in this paper, the study results were calculated by applying the method proposed in this study only for the two observation periods (IPP-1 and IPP-3) among four periods of Birch and Pine pollen. How about showing the results by applying this method for the period of IPP-2 and IPP-4? In this case, not only Birth and Pine, but also other types of pollen or a mixture of various types, couldn't we derive meaningful research results?*
We agree that such investigation of other types of pollen or a mixture of various types are very important. In our investigation, we also applied the method for IPP-2 and IPP-4, but no good results were found. This is explained in the manuscript (Page 11 lines 14-17):

*The retrieval of depolarization ratios for pure spruce or pure nettle pollen was not possible with this dataset. During IPP-2, there was always a mixture of birch and spruce pollen with variable mixing ratio; in addition, the number of available measurements is limited. For nettle pollen, we have observed relatively small depolarization ratio values, together with a small variation, which makes the separation more challenging.*

We are working on this and we are collecting more data to be able to reveal the properties for different pure pollen types.

---

## Author Comment (AC3) · 25 Oct 2020

**Response to Referee #1**

**For referee comment no.2 (RC3: 'clarification')**

Thank you for carefully reading the manuscript and providing useful suggestions to improve the paper. The replies to the referee comments are given below. The referee comments are highlighted in blue, and numbering with C*n*. Our responses are in black. The sentences in the manuscript are between the quotation marks, with the modifications in the revised manuscript in red.

**C1 Thank you for your quick reply to my Referee comment. I now have a better picture of what you are doing. I also believe that your presentation is unnecessarily convoluted. The manuscript should be simplified in the presentation of the method as well as with regard to the used language and parameters. For instance, it would be much easier to follow your reasoning if you were to use the Ångtröm exponent Å in your method rather than the unphysical parameters $\eta$, $\eta'$, $\eta''$, and $\hat{\eta}$. I had to continuously go back and forth to remind myself what all those parameters represent. I also don't agree with your reference to simulations, a direct model, and an indirect model. What your present is a purely analytical treatment of synthetic and measured lidar profiles.**

Thank you for the suggestion. We have simplified the presentation of the method, and separated the presentation of the methodology from the presentation of the results (following suggestion C5). We add descriptions after the "direct model" and "inverse model" as following to clarify the manuscript. The structure of revised manuscript has been changed as:
"

   1 Introduction

   2 Site and instruments

   3 Methodology – a synthetic simulator

      3.1 Direct model – generation of synthetic optical profiles

      3.2 Inverse model – retrieval of depolarization ratio

      3.3 Uncertainty study

   4 Results

      4.1 Pollen grain and intense pollination period

      4.2 Optical properties of pollen layer

          4.2.1 Pollen layer

          4.2.2 Lidar-derived optical properties

      4.3 Estimation of optical properties for pure pollen from lidar observations

          4.3.1 Pollen optical properties at 532 nm

          4.3.2 Pollen optical properties at 1064 nm and 355 nm

   5 Summary and conclusions
"

However, we use the parameter $\eta$ (a function of Ångtröm exponent) because this parameter and the pollen backscatter contribution have a linear relationship. As there is a power law relationship between the Ångtröm exponent and the pollen backscatter contribution, introducing the parameter $\eta$ simplifies the calculation.

This was not clearly stated in the manuscript and is improved in the revised version of the manuscript. The related equation is added in the revised manuscript and detailed calculations are added in the supplement. The methodology section is modified to make it easier to follow. In the revised version we include a Table (Table 2) to clearly present the parameters $\eta$. We only keep 2 parameters $\eta$, $\eta'$ in the revised manuscript, because $\eta''$(in the original version) is equal to $1/\eta$, so $\frac{1}{\eta}$ is used for the description of depolarization ratio at 355 nm.

So in the section 3.1 of revised version, we have:

"

Pollen backscatter contribution, denoted as $\chi_{\text{pollen}}$ (Eq.4), is defined as the ratio of pollen backscatter coefficient ($\beta_{\text{pollen}}$) and the total particle backscatter coefficient ($\beta_{\text{particle}}$). Note that the use of "particle" here is to distinguish from "molecular".

$$\chi_{\text{pollen}}(\lambda, z) = \frac{\beta_{\text{pollen}}(\lambda,z)}{\beta_{\text{particle}}(\lambda,z)} \tag{4}$$

We investigate here the relationship of the backscatter-related Ångström exponent of total particles ($\text{Å}_{\text{particle}}$) and pollen backscatter contribution ($\chi_{\text{pollen}}$) at different wavelengths (the detailed calculation is given in the supplement), resulting a power law relationship:

$$\frac{\lambda_1}{\lambda_2}^{-\text{Å}_{\text{particle}}(\lambda_1,\lambda_2)} = \left(\frac{\lambda_1}{\lambda_2}^{-\text{Å}_{\text{pollen}}(\lambda_1,\lambda_2)} - \frac{\lambda_1}{\lambda_2}^{-\text{Å}_{\text{background}}(\lambda_1,\lambda_2)}\right)\chi_{\text{pollen}}(\lambda_2) + \frac{\lambda_1}{\lambda_2}^{-\text{Å}_{\text{background}}(\lambda_1,\lambda_2)} \tag{5}$$

The wavelength pairs ($\lambda_1, \lambda_2$) are selected as (355,532), (532,355), or (1064,532) in this study. In order to simplify the calculation, we introduce two parameters η, and η' as a function of the backscatter-related Ångström exponent between 355 and 532 nm or between 532 and 1064 nm, for the total particle backscatter coefficients:

$$\begin{cases} \eta = \left(\frac{355}{532}\right)^{-\text{Å}_{\text{particle}}(355,532)} \\ \eta' = \left(\frac{1064}{532}\right)^{-\text{Å}_{\text{particle}}(1064,532)} \end{cases} \tag{6}$$

The pairs of parameter η or η' and $\chi_{\text{pollen}}$ at different wavelengths resulting linear relationships are reported in Table 2. For example, the pollen backscatter contribution at 532 nm ($\chi_{\text{pollen}}(532)$) is inversely proportional to the parameter η. Using the previous 6 simulated cases, a perfect linear relationship is found to fit the η versus $\chi_{\text{pollen}}(532)$ (Fig.2).

Table 2. The pairs of the parameter η($\text{Å}_{\text{particle}}$) and $\chi_{\text{pollen}}$ at different wavelengths resulting linear relationships are reported.

| Wavelength pair($\lambda_1, \lambda_2$) [nm] | Pollen backscatter contribution at $\lambda_2$ | Backscatter-related Ångström exponent $\text{Å}(\lambda_1, \lambda_2)$ | Parameter of $\text{Å}(\lambda_1, \lambda_2)$, linearly correlating with $\chi_{\text{pollen}}$ | Formulate |
|---|---|---|---|---|
| $\lambda_1$=355 $\lambda_2$=532 | $\chi_{\text{pollen}}(532)$ | $\text{Å}_{\text{particle}}(355,532)$ | η | $\eta = \left(\frac{355}{532}\right)^{-\text{Å}_{\text{particle}}(355,532)}$ |
| $\lambda_1$=532 $\lambda_2$=355 | $\chi_{\text{pollen}}(355)$ | $\text{Å}_{\text{particle}}(532,355)$ | $\frac{1}{\eta}$ | $\frac{1}{\eta} = \left(\frac{532}{355}\right)^{-\text{Å}_{\text{particle}}(355,532)}$ |
| $\lambda_1$=1064 $\lambda_2$=532 | $\chi_{\text{pollen}}(532)$ | $\text{Å}_{\text{particle}}(1064,532)$ | η' | $\eta' = \left(\frac{1064}{532}\right)^{-\text{Å}_{\text{particle}}(532,1064)}$ |

"

In the section 3.2 of revised version, we modified as:

"

The only remaining unknown to solve the Eq.7 is the depolarization ratio for pure pollen ($\delta_{\text{pollen}}$). Next we use previously simulated $\beta_{\text{particle}}$ and $\delta_{\text{particle}}$, and the assumed $\delta_{\text{background}}$. From now on, 532 nm will be the default wavelength (if not otherwise specified). The wavelength pair ($\lambda_1, \lambda_2$) is selected as (355,532) in this section. Mean values of optical properties inside the pollen layer are considered in this study; it is also possible to use values of each bin of the synthetic profile which will lead to the same conclusion. Mean values of backscatter-related Ångström exponent between 355 and 532 nm inside the pollen layer, denoted as Å(355,532), can be easily retrieved.

"

In the section 4.3.2 of revised version, for study at 1064 nm, we modified as:

"

Similar study was performed to investigate the relationship between backscatter-related Ångström exponent between 532 and 1064 nm (Å(1064,532)) and pollen backscatter contribution at 532 nm,

here we use another parameter $\eta'$ (Eq.6), which is a function of $\text{Å}(1064,532)$, for the total particle backscattering. From the earlier simulations, we found out that the pollen backscatter contribution at 532 nm ($\chi_{\text{pollen}}(532)$) is proportional to the parameter $\eta'$, considering the Eq.5 using the wavelength pair of $\lambda_1$=1064 and $\lambda_2$=532.
"

In the section 4.3.2 of revised version, for study at 355 nm, we modified as:
"

The inverse model was applied here for the backscatter-related Ångström exponent between 355 and 532 nm ($\text{Å}(532,355)$) and pollen backscatter contribution at 355 nm, using a third parameter $\frac{1}{\eta}$ (as in Eq.6, a function of $\text{Å}(532,355)$), which is proportional to the pollen backscatter contribution at 355 nm, considering the Eq.5 using the wavelength pair of $\lambda_1$=532 and $\lambda_2$=355.
"

**Here is a description of your method as I understand it:**

**You are defining a set of synthetic lidar profiled for pollen and background aerosol using the set profile shape and the parameters in Table 3. These profiles are then combined to obtain a profile for the mixture of the two. You use Eq. (S5), which is the same as Eq. (13) in Tesche et al. (2009), to get the particle linear depolarisation ratio of the mixture. Your Eq. (S5) is transformed to Eq. (6) / Eq. (14) in Tesche et al. (2009) by substituting $\beta_{pollen} = \beta_{total} - \beta_{background}$. You now have full knowledge of the system and can calculate the pollen ratio $\chi = \beta_{pollen}/\beta_{total}$. Finally, you show that the relationship between Å and $\chi$ can also be analytically described to find the value of Å related to $\chi$ = 1. Per definition in Table 3, Å is zero for $\chi$ = 1 in your synthetic data.**

**Comments:**

**C2 • I understand that your choice of parameters in Table 3 is not critical for presenting the overall approach. Nevertheless, it would be nice to get an idea of why those specific values have been selected. In particular, I find the background Ångtröm exponent of 3 quite large.**

**• Note that I am using the Ångtröm exponent in my description as I find it much easier to follow the steps using a parameter that bears physical meaning.**

You are right, the choice of parameters in Table 3 (Table 1 in the revised version) is not critical for presenting the method. There are descriptions on the choices of parameters in the 1st paragraph in section 3.3.1 of the old version of manuscript. We have made some modifications in the revised version to make it more clear.
We have also changed the assumption value for non-pollen particle Ångtröm exponent as 2 (instead of 3) in the revised version. This value of 2 is more realistic. Thank you for pointing this out.

We add information in section 3.1 as:
"

The values are based on our lidar measurements (Bohlmann et al., 2019) or literature (e.g. Illingworth et al., 2015). The *background* here refers to non-depolarizing background aerosols (non-pollen particles), which can be polluted continental or biomass burning aerosols. The depolarization ratio at both 355 and 532 nm of non-pollen particle ($\delta_{\text{background}}$) are selected as 0.03, which is a mean value for pollen-free periods at our measurement site. Bohlmann et al. (2019) shows that the pollen can generate strong depolarization, thus the depolarization ratio at 532 nm of pure pollen particle ($\delta_{\text{pollen}}$) are selected as 0.35 as the initial value for the simulation in this section. Pollen grains are quite big and thus can be assumed to be wavelength independent on the backscatter at wavelengths of 355 nm and 532 nm, with the backscatter-related Ångström exponent ($\text{Å}_{\text{pollen}}$) of 0. The backscatter-related

Ångström exponent between 355 and 532 nm of non-pollen particle ($Å_{background}$) is assumed to be 2, regarding the previous studies over Arctic regions (e.g. Schmeisser et al., 2018; Tomasi et al., 2012).

”

We have also modified in the first paragraph of section 3.1 as:

“

The optical and physical parameters used in the direct calculation are presented in Table 1; these parameters are named as "initial values" for the simulation.

…

In addition, the conclusion of the simulation section is not depended on the assumed profile shape or height; and the initial values are not critical for presenting the overall approach.

”

**C3** • **You probably don't even need to include the vertical integration of $\chi(z)$. If you use the profile of $\chi(z)$ from your synthetic data, they will still line up perfectly. In the application to real-life measurements, you might also want to leave out the vertical integral as this would require initial knowledge of the pollen layer extend in the measurements. I have a feeling that values of $\chi(z)$ outside the pollen layer will be easy to recognise and screen out in the display of Å over $\chi$.**

We agree, and change the equation as Eq.4 in our reply to the comment C1. We also add description on this in section 3.2 of revised version as:

“

Mean values of optical properties inside the pollen layer are considered in this study; it is also possible to use values of each bin of synthetic profile which will lead to the same conclusion.

”

For synthetic simulation it is the same, but for a real measurement, the use of pollen layer is preferable. We decide to keep the use of the mean values of pollen layer in this study, because it can increase the signal to noise ratio (SNR), and also eliminate the impact of other possible lofted aerosol.

**Next steps in the methodology:**

**C4** **Now that you know everything about your model aerosol, you basically turn around and use the same set of equations in the other direction with $\delta_{pollen}$ as the unknown parameter. I would expect an inverse model to be completely independent from the earlier calculations. Instead, your just re-shuffle the equations used before, vary the input value of $\delta_{pollen}$, and iterate until you have found the value of $\delta_{pollen}$ for which Å = 0 at $\chi$ = 1. It's as simple as that but it took me quite a while to get there based on your description. I'd therefore encourage you to simplify the presentation of your methodology.**

Thank you for the suggestion. We have simplified the presentation of methodology. In the section 3.2 of revised version, we changed the flow chart as following and modified the text as:

“

Mathematically, the depolarization ratio for pure pollen can be calculated using Eqs.4,5,7, as other variables are known or can be assumed. Nevertheless, we developed a retrieval method for this inverse model, so that it can be easier applied to the real lidar measurements, especially for investigating the depolarization ratio with different values of the unknown $Å_{pollen}$. An iterate approach is used. In the first step, the depolarization ratio for pure pollen was assumed to be several different values (within the range between 0.03 to 1), denoted as $\delta_x$, in the simulator. Related pollen backscatter contribution ($\chi_{pollen}(532)$) inside the pollen layer, can be retrieved using Eqs.4 and 7. As its value depends on the assumed pollen depolarization ratio ($\delta_x$), it can be expressed as $\chi_{pollen}(\delta_x, 532)$.

The relationship of Å(355,532) and $\chi_{\text{pollen}}(\delta_x, 532)$ was investigated using the parameter η (Eqs.5 and 6. Examples of scatter plots using mean values of η and $\chi_{\text{pollen}}(\delta_x, 532)$ in the pollen layer for cases under the assumptions of $\delta_x$ =0.1, 0.2, 0.3, 0.4 and 0.5 are shown in Fig.3. For these relationships, perfect linear fits (linear regression relationship) can be found and plotted as dotted lines in the Fig.3, following the simplified equation from Eqs.5 and 6:

$$\eta(\chi_{\text{pollen}}(\delta_x, 532)) = a_1 \cdot \chi_{\text{pollen}}(\delta_x, 532) + a_0 \tag{8}$$

The fitting coefficient $(a_1, a_0)$ values to determine the estimated parameter η are defined as in Eq.5. Until this step of the inverse model, no assumption on the $\text{Å}_{\text{pollen}}$ was made, thus $a_1$ varies for different assumed values of $\delta_x$. But $a_0$ is constant as the $\text{Å}_{\text{background}}$ is known. Theoretically, for each linear fit equation, $\chi_{\text{pollen}}(\delta_x, 532)$ values can range from 0 to 1, with 0 meaning no pollen and 1 meaning 100 % pollen in the observed aerosol particle population. Therefore, for each assumed $\delta_x$, the η value for $\chi_{\text{pollen}}(\delta_x, 532)=1$ can be defined as the value for the pure pollen, and denote as $\eta_{\text{pure}}(\delta_x, 532)$.

In Sect. 3.1, the initial value of the backscatter-related Ångström exponent between 355 and 532 nm of pure pollen (denoted as $\text{Å}_{\text{pollen}}$) is 0, which results in an initial value of 1 for the parameter η. In this simulation, we assumed that the same value ($\widehat{\text{Å}}_{\text{pollen}}=0$) should be retrieved; the goal was thus to find the value of 1 for $\eta_{\text{pure}}$. From previous results shown in Fig.3, we can see a $\delta_x$ between 0.3 to 0.4 may result in a $\eta_{\text{pure}}=1$ (the black triangle in Fig.3).

Hence, in the second step, more $\delta_x$ values between that range (0.3 – 0.4) were used in the simulation, and one can retrieve the relative value of $\eta_{\text{pure}}(\delta_x, 532)$ for each case. These values are presented in Fig.4. The relationship between $\delta_x$ and $\eta_{\text{pure}}(\delta_x, 532)$ is not perfectly linear, but for these data inside the considered range, a good linear fit can be found with high correlation coefficients ~-1. As there is noise in real lidar measured profiles, two or more values of $\delta_x$ may be found as good solutions. However, after we introduce this additional second linear fit, only one solution will be retrieved in the end.

[Figure]

**Figure 5. Flow chart of the inverse model for the retrieval of depolarization ratio value for pure pollen. The orange boxes are for the measured parameters (or simulated output from the direct model), blue boxes for the assumptions/manual input and the green boxes for the estimations/calculations. Detail description is in Sect. 3.2. The wavelength pair $(\lambda_1, \lambda_2)$ is selected as (355,532), (532,355), or (1064,532) in this study.**

„

**Now some more comments regarding the manuscript:**

**C5 • I again strongly encourage you to separate the presentation of the methodology from the presentation of the results. This is customary in scientific writing and allows the reader not only to better follow your reasoning but also to separate more general relations from your specific results.**

We agree, and change the structure as in reply to the comment C1.

**C6 • I am quite sceptical about Section 3.4.2. You have defined no profile of $\beta_{1064}$ and no $\mathring{A}_{532=1064}$ in your synthetic data set. How could you know how to interpret your findings when applying this extended approach to real-life data? Your retrieval of $\delta_{pollen}$ at 355 nm is basically analogous to that at 532 nm. In fact, the choice of values in Table 3 indicates that profiles at 355 and 532 nm should be identical. Why not use the same method at 355 and 532 nm? This should already be discussed in the theory section.**

As in our reply to the comment C1, there is a linear relationship (Eq.5) between $\frac{\lambda_1}{\lambda_2}^{-\mathring{A}_{particle}(\lambda_1,\lambda_2)}$ and $\chi_{pollen}(\lambda_2)$.

For 1064 nm study, we use $\lambda_1$=1064 and $\lambda_2$=532, so $\eta' = \left(\frac{1064}{532}\right)^{-\mathring{A}_{particle}(532,1064)}$ with $\chi_{pollen}(532)$ as parameters for the linear relationship. Thus using the pollen backscatter contribution at 532 nm ($\chi_{pollen}(532)$), we estimate the backscatter-related Ångström exponent between 532 and 1064 nm $\mathring{A}_{particle}(532,1064)$.

For 355 nm study, we need to use $\chi_{pollen}(355)$ instead of $\chi_{pollen}(532)$, thus the wavelength pair should be $\lambda_1$=532 and $\lambda_2$=355 instead of $\lambda_1$=355 and $\lambda_2$=532, then the parameter should be $\frac{1}{\eta}$ instead of $\eta$. In our previous version of manuscript we used a 3rd parameter $\eta''$, but we changed it to $\frac{1}{\eta}$ for the revised manuscript.

More details are given in our reply to the comment C1.
In addition, in the revised version of supplement, we have added a section for the detailed calculation for Eq.5:

**2 Relationship of $\mathring{A}_{particle}$ and $\chi_{pollen}$ (Eq.5 in the manuscript)**

Two aerosol populations, pollen (depolarizing) and background (non-depolarizing) aerosols are considered. The backscatter coefficient of the total particles is the sum of backscatter coefficient of both pollen and background aerosols:

$$\beta_{particle}(\lambda_1) = \beta_{pollen}(\lambda_1) + \beta_{background}(\lambda_1) \tag{S7a}$$
$$\beta_{particle}(\lambda_2) = \beta_{pollen}(\lambda_2) + \beta_{background}(\lambda_2) \tag{S7b}$$

Similar as Eq.2 in the manuscript, the backscatter-related Ångström exponent (Å) can also be expressed in this equation:

$$\frac{\lambda_1}{\lambda_2}^{-\mathring{A}_x(\lambda_1,\lambda_2)} = \frac{\beta_x(\lambda_1)}{\beta_x(\lambda_2)} \tag{S8}$$

The index *x=pollen, background* or *particle* denotes the backscatter-related Ångström exponent of pollen, background or total particles.

We replace the top part of right side of Eq.S8 with *x= particle* with Eq.S7. And further use expression of $\beta_{pollen}(\lambda_2)$ and $\beta_{background}(\lambda_2)$ to replace the $\beta_{pollen}(\lambda_1)$ and $\beta_{background}(\lambda_1)$ in Eq.S7a, based on Eq.S8. Thus we have:

$$\frac{\lambda_1}{\lambda_2}^{-\mathring{A}_{particle}(\lambda_1,\lambda_2)} = \frac{\frac{\lambda_1}{\lambda_2}^{-\mathring{A}_{pollen}(\lambda_1,\lambda_2)}*\beta_{pollen}(\lambda_2)+\frac{\lambda_1}{\lambda_2}^{-\mathring{A}_{background}(\lambda_1,\lambda_2)}*\beta_{background}(\lambda_2)}{\beta_{particle}(\lambda_2)} \tag{S9}$$

After replacing $\beta_{background}(\lambda_2)$ with Eq.S7b, the equation can be expressed as:

$$\frac{\lambda_1}{\lambda_2}^{-\text{Å}_{\text{particle}}(\lambda_1,\lambda_2)} = \frac{\left(\frac{\lambda_1}{\lambda_2}^{-\text{Å}_{\text{pollen}}(\lambda_1,\lambda_2)} - \frac{\lambda_1}{\lambda_2}^{-\text{Å}_{\text{background}}(\lambda_1,\lambda_2)}\right)*\beta_{\text{pollen}}(\lambda_2) + \frac{\lambda_1}{\lambda_2}^{-\text{Å}_{\text{background}}(\lambda_1,\lambda_2)}*\beta_{\text{particle}}(\lambda_2)}{\beta_{\text{particle}}(\lambda_2)} \quad \text{(S10)}$$

Using the definition of pollen backscatter contribution (Eq.4 in the manuscript), a linear relationship between $\frac{\lambda_1}{\lambda_2}^{-\text{Å}_{\text{particle}}(\lambda_1,\lambda_2)}$ and $\chi_{\text{pollen}}(\lambda_2)$ can be retrieved for the wavelength pair $(\lambda_1, \lambda_2)$:

$$\frac{\lambda_1}{\lambda_2}^{-\text{Å}_{\text{particle}}(\lambda_1,\lambda_2)} = \left(\frac{\lambda_1}{\lambda_2}^{-\text{Å}_{\text{pollen}}(\lambda_1,\lambda_2)} - \frac{\lambda_1}{\lambda_2}^{-\text{Å}_{\text{background}}(\lambda_1,\lambda_2)}\right)\chi_{\text{pollen}}(\lambda_2) + \frac{\lambda_1}{\lambda_2}^{-\text{Å}_{\text{background}}(\lambda_1,\lambda_2)} \quad \text{(S11a)}$$

A similar formulate is found for the wavelength pair $(\lambda_2, \lambda_1)$ when considering $\chi_{\text{pollen}}(\lambda_1)$:

$$\frac{\lambda_2}{\lambda_1}^{-\text{Å}_{\text{particle}}(\lambda_1,\lambda_2)} = \left(\frac{\lambda_2}{\lambda_1}^{-\text{Å}_{\text{pollen}}(\lambda_1,\lambda_2)} - \frac{\lambda_2}{\lambda_1}^{-\text{Å}_{\text{background}}(\lambda_1,\lambda_2)}\right)\chi_{\text{pollen}}(\lambda_1) + \frac{\lambda_2}{\lambda_1}^{-\text{Å}_{\text{background}}(\lambda_1,\lambda_2)} \quad \text{(S11b)}$$

**C7 • While the information on pollen type and concentration in Figures 11, S4, S5, and S8 is certainly good to have, it is not needed in those plots. Instead, they distract from the intended message. As stated above, I'd expect that the display would work just the same using all values of $\chi(z)$. The ones outside the pollen layer should be easy to identify as (strong?) deviations from the desired relationship. Using the integrated parameter with the actual measurements might reduce signal noise. However, it requires knowledge of the base and top of the pollen layer as you don't want to include values of $\chi(z)$ outside of this layer in your integration.**

We think these figure show data from real measurements, so it is good to present. Please also check our reply to the comment C3 for the reason of using pollen layer.

We have removed the fig.11 (in the original version of manuscript), and add a fig.12 (in the revised version). These 2 figures are similar, but in the new fig.12 we have applied the retrieved pollen depolarization ratio value, i.e. 0.24 instead of 0.2 for fig.12(a), 0.36 instead of 0.4 for fig.12(b).

**C8 • You might want to state that this method can also be applied to other aerosol mixtures to retrieve the particle linear depolarisation ratio related to aerosol types that are dominated by coarse particles (Å355=532 = 0 needs to be fulfilled), as long as the particle linear depolarisation ratio of the second aerosol types is known or can be reasonably well approximated. An obvious application would be the retrieval of the particle linear depolarisation ratio related to undiluted mineral dust from different source regions. The lidar measurements for such a retrieval could be performed further away from the source regions, which translates into a strong reduction of logistical effort.**

Thank you for the suggestion. We had mentioned such application in the end of section 3.3.2 of old version of manuscript, and we have modified it in Sect. 3.2 of the revised version:
"

This method can also be applied to other two aerosol types (e.g., dust and non-dust aerosols), under the condition that the depolarization ratio of one aerosol type is the only unknown parameter, and other parameters are known or can be assumed, as long as both the depolarization ratio and the backscatter-related Ångström exponent of the two aerosol types are different.
"

We have also added such information in the conclusion. At the end of the conclusion of the revised version, we have added:
"

This method can also be applied to other aerosol mixtures (e.g., dust and non-dust aerosols) to retrieve the particle linear depolarization ratio related to aerosol types, under the condition that the depolarization ratio of one aerosol type is the only unknown parameter, and other parameters are known or can be reasonably well approximated. Note that the two constrains mentioned in Sect.3.1 should be considered: both the depolarization ratio and the backscatter-related Ångström exponent of the two aerosol types should be different.
"

---

## Author Comment (AC4) · 25 Oct 2020

**Response to Referee #3**

Thank you for carefully reading the manuscript and providing useful suggestions to improve the paper. The replies to the referee comments are given below. The referee comments are highlighted in blue with our responses in black. The sentences in the manuscript are between the quotation marks, with the modifications in the revised manuscript in red.

**The authors present multi-wavelength Raman polarization lidar measurements of pollen layers in Finland combined with a Burkard pollen sampler. Active remote sensing measurements of pollen are rarely found in literature. Therefore, the present manuscript enriches our knowledge about the optical properties of abundant pollen types such as birch and pine pollen. Northern Europe (Finland) is a good location for such a study as it is less affected by other depolarizing aerosol particles such as mineral dust. Additionally, the authors present a novel approach to derive the depolarization ratio of pure pollen layers. Although it is related to some uncertainties, it is a big step forward compared to just presenting the layer mean values. I support the idea that measurements of the depolarization ratio at various wavelengths should be enforced in future pollen-related studies. Polarization lidars may in future support pollen forecasts and help citizens with pollen allergy thanks to the characterization of pure pollen types by these authors. The quality of the figures and tables is high.**
**Finally, I recommend publication after minor revisions.**

**Major remarks:**

**1. You use a value of 3 for the backscatter-related Ångström exponent of the background aerosol. Do you have any statistical evidence of this value for the station at Kuopio? Is it a mean value for the pollen-free periods? And how sensitive is your analysis to this assumption?**

Thank you for pointing it out. The choice of parameters in Table 3 (Table 1 in the revised version) for the simulation is not critical for presenting the overall approach.
We have changed the assumption value for non-pollen particle Ångtröm exponent ($Å_{background}$) as 2 (instead of 3) in the revised version. This value of 2 is more realistic. We have changed all the related results and figures. The assumption for $Å_{background}$ is only used in the simulation part, and is not considered for the pollen depolarization ratio retrieval, so the actual results using lidar measurements will not change. We have made some modifications for the revised version to make is more clear.

We have added information in section 3.1 as:
"
The optical and physical parameters used in the direct calculation are presented in Table 1; these parameters are named as "initial values" for the simulation. The values are based on our lidar measurements (Bohlmann et al., 2019) or literature (e.g. Illingworth et al., 2015). The *background* here refers to non-depolarizing background aerosols (non-pollen particles), which can be polluted continental or biomass burning aerosols. The depolarization ratio at both 355 and 532 nm of non-pollen particle ($\delta_{background}$) are selected as 0.03, which is a mean value for pollen-free periods at our measurement site. Bohlmann et al. (2019) shows that the pollen can generate strong depolarization, thus the depolarization ratio at 532 nm of pure pollen particle ($\delta_{pollen}$) are selected as 0.35 as the initial value for the simulation in this section. Pollen grains are quite big and thus can be assumed to be wavelength independent on the backscatter at wavelengths of 355 nm and 532 nm, with the backscatter-related Ångström exponent ($Å_{pollen}$) of 0. The backscatter-related Ångström exponent between 355 and 532 nm of non-pollen particle ($Å_{background}$) is assumed to be 2, regarding the previous studies over Arctic regions (e.g. Schmeisser et al., 2018; Tomasi et al., 2012). Note that these values can be changed freely for the simulation under 2 constraints: i. depolarization ratio of pollen (depolarizing one) should be higher than the depolarization ratio of background aerosol (nondepolarizing one), ii. the values of backscatter-related Ångström exponent for pollen and non-pollen particle should be different. In addition, the conclusion of the simulation section is not depended on the assumed profile shape or height; and the initial values are not critical for presenting the overall approach.

"

We have also investigated the sensitivity of this assumption in the simulation section. For the uncertainty study due to initial and assumed Ångström exponent in section 3.3 of the revised version, we have modified as:

"

In the presented cases, we assumed that the backscatter-related Ångström exponent between 355 and 532 nm of pure pollen to be used in the inverse model (denoted as $\widehat{\text{Å}}_{\text{pollen}}$) is 0, which was the same as the initial value ($\text{Å}_{\text{pollen}}$) of direct model. But in the reality, such information is not always available. Under different initial values of $\text{Å}_{\text{pollen}}$, there will be a bias on the estimated values of pollen depolarization ratio if the assumed value is different (i.e. $\widehat{\text{Å}}_{\text{pollen}} \neq \text{Å}_{\text{pollen}}$). For example, if the initial value $\text{Å}_{\text{pollen}}$ is 0.25 (i.e. $\eta_{\text{pure}}$=1.11), but we keep the assumption of $\widehat{\text{Å}}_{\text{pollen}}$=0 in the inverse model, the estimated pollen depolarization ratio is found to be 0.39 with a bias of 0.04 (show in Fig. S3 in the supplement). The uncertainty due to the difference between the initial value of $\text{Å}_{\text{pollen}}$ and assumed $\widehat{\text{Å}}_{\text{pollen}}$ were simulated (show in Fig. S4 in the supplement), where $\widehat{\text{Å}}_{\text{pollen}}$ is always assumed as 0 in the inverse model. For initial values of $\text{Å}_{\text{pollen}}$=±0.5 (i.e. bias of 0.5 on the assumed value of 0), relative uncertainties were assessed as ~30 %. This uncertainty due to the difference of initial values of $\text{Å}_{\text{pollen}}$ and $\text{Å}_{\text{background}}$ was also investigated. The larger the difference between two values ($\text{Å}_{\text{background}} - \text{Å}_{\text{pollen}}$), the smaller the uncertainty. For instance, if we use 3 (instead of 2) as the initial value of $\text{Å}_{\text{background}}$, the estimated pollen depolarization ratio is 0.37 (instead of 0.39) with a smaller bias for the above example.

"

In addition, we can retrieve the non-pollen particle Ångtröm exponent using our lidar measurements, based on the presented algorithm (using Eq.5 in the revised version). We found $\text{Å}_{\text{background}}$ values of 2.0 and 1.9 for IPP-1 and IPP-3, respectively. These results are added in section 4.3.1 in the revised version:

"

Under the assumption that the backscatter-related Ångström exponent between 355 and 532 nm of pure pollen (denoted as $\text{Å}_{\text{pollen}}$) is 0 (i.e. $\eta_{\text{pure}}$=1), depolarization ratio of 0.24 or 0.36 were found for IPP-1 or IPP-3, respectively, which are related to the pure birch or pure pine pollen (Table 5). The scatter plots of mean $\eta$ and $\chi_{\text{pollen}}(\delta_x, 532)$ are shown in Fig. 12: (a) for IPP-1 with the pollen depolarization ratio of 0.24, and (b) for IPP-3 with the pollen depolarization ratio of 0.36. Good linear regression relationships are found for both cases, and two things should be highlighted: (1) $\text{Å}_{\text{pollen}}$ is 0 (i.e. $\eta_{\text{pure}}$=1) for 100 % pollen in the observed aerosol particle population (i.e. $\chi_{\text{pollen}}$=1); (2) without pollen in the air (i.e. $\chi_{\text{pollen}}$=0), the backscatter-related Ångström exponent between 355 and 532 nm of non-pollen particles ($\text{Å}_{\text{background}}$) can be calculated, resulting values of 2.0 for IPP-1 and 1.9 for IPP-3 (i.e. $\eta$ of 2.28 for IPP-1, 2.18 for IPP-3).

[Figure]

Figure 12. Mean values of the parameter η against pollen backscatter contribution at 532 nm ($\chi_{\text{pollen}}(\delta_x, 532)$) inside the pollen layers, during the IPP-1 (a) and IPP-3 (b). η is a parameter using backscatter-related Ångström exponent between 355 and 532 nm (Eq.6). The pollen depolarization ratio $\delta_x$ at 532 nm is assumed to be 0.24 for (a) or 0.36 for (b). Linear regression lines are drawn by dotted lines, with fitting equation shown (Eq.5 or 8). The correlation coefficient ($R^2$) is also given. The size denotes the total pollen concentrations measured by the Burkard sampler on roof level; the colour represents the number concentration of the dominant pollen (a: birch, b: pine) against the total pollen number concentration. Similar figures using different assumed values of pollen depolarization ratio can be found in Fig. S5 and Fig. S6 in the supplement.
"

**2. Your novel approach for getting the depolarization ratio of the pure aerosol type is remarkable. I am just wondering whether the mixture of continental background aerosol and pollen has a significant effect on the lidar ratio, too. It would be great to have the lidar ratio and the depolarization ratio for pure birch and pine pollen at the end. Please comment on this.**

We think the pollen has effect on the lidar ratio. But more nighttime measurements (for lidar ratio retrieval) for intense pollination cases are need for investigating such scientific question. If we find a case where there is a pollen layer in the free troposphere (without contamination of aerosols in PBL), with strong depolarization ratio and small Ångström exponent, it will be good to study the lidar ratio for such layer to retrieve "pure values".

**Minor remarks:**

**3. P5,L25: "The extinction-related and backscatter-related Ångström exponent were also retrieved for pollen layers." – Is the extinction-related Ångström exponent shown somewhere? It must not be shown in the manuscript, some descriptive words are sufficient.**
We have retrieved the extinction-related Ångström exponent, but we haven't presented such parameter as the available data are limited. We have added descriptions as:
"

The extinction-related (not shown in this study) and backscatter-related Ångström exponent were also retrieved for pollen layers.
"

**4. P10,L30 The Ångström exponent is related to extinction or backscatter?**
Thank you for pointing this out. It is the extinction-related, we have added information in the revised version.
"

For big particles as dust, Mamouri and Ansmann (2014) reported extinction-related Ångström exponent between 440 and 675 nm with values of -0.2 for coarse dust and 0.25 for total dust.
"

**5. P11,L10 Are the measurements presented by Cao et al., (2010) performed at exactly 180_ backscatter direction? This is not so easy to achieve in chamber experiments. Maybe there is an additional source for the discrepancy arising from the optical design of the Cao measurements?**

Cao et al. (2010) performed the measurement at 180 deg direction. The lidar measurements were made in an aerosol chamber located 100 m away from the lidar. 2 g of the selected pollen is disseminated within a few seconds with a pneumatic nozzle in the chamber.

They have pointed out that "the reported values are not exempt from specificities regarding the experiments as they were conducted" and have a discussion on this aspect. For example, the dissemination device used has an influence on the amount of agglomeration of the particles, and that certainly could affect the depolarization ratios. RH can be another reason, as for their experiment dry aerosols are being dispersed, whereas in our study, we focus on the aerosols in the atmosphere.

As we mentioned in the manuscript:

"

These values are higher than what we retrieved in this study, but it has to be kept in mind that these two experiments have been conducted in quite different environments and conditions.

"

**6. Fig. 1+2 and Tab. 1: Please provide the year (2016) whenever you provide dates. Do it in the caption or just like this "Date mm/dd in 2016 [UTC]".**

Thank you for your comment. The corrections have been done.

**7. How do you get to the uncertainty range +/-5% for pine pollen? Varying the Ångström exponent by +/- 0.5 leads to values of 26 to 44% (Fig. 12 and P11,L9).**

Thank you for pointing it out, we agree that it was confusion and not correct. We made the corrections to make it more clear in the revised version. Please also check our reply to the comment "Major remarks 1." for the uncertainty study for the simulation section.

For the uncertainty study of the real lidar measurements, we modified as:

"

Uncertainty study was investigated based on method describe in Sect.3.3 using a Monte Carlo approach. The overall relative uncertainties of the lidar-derived backscatter coefficients are of the order of 5 %–10 % (Baars et al., 2012), we took 10 % here in the simulation. Initial pollen depolarization ratio values were selected as 0.24 for birch and 0.36 for pine for the uncertainty simulation; initial backscatter-related Ångström exponent between 355 and 532 nm of non-pollen particles were selected as 2.0 and 1.9 for IPP-1 and IPP-3, respectively. Based on the lidar observations (Fig. 12), the simulated cases were selected so that the $\chi_{pollen}$ values range from 2 % to 60 % for birch and 2 % to 90 % for pine. The initial input $\mathring{A}_{pollen}$ in the direct model and assumed $\hat{\mathring{A}}_{pollen}$ in the inverse mode were both selected as 0. Estimated uncertainties were found as 2.4 % for birch and 2.9 % for pine (Table 5). Note that the different initial input values of $\mathring{A}_{pollen}$ may introduce important additional bias. If we assume the true value of $\mathring{A}_{pollen}$ is between -0.5 to 0.5 (i.e. values of $\eta_{pure}$ from 0.82 to 1.22, shown by red dotted lines in Fig. 11), depolarization ratios of 0.19 to 0.27 can be found for birch pollen, and 0.26 to 0.44 can be found for pine pollen.

Table 5. Linear depolarization ratios for pure pollen. The assumption of backscatter-related Ångström exponent between 355 and 532 nm for pollen should be 0 was applied for this study. The uncertainty on backscatter-related Ångström exponent of pollen was not taken into account for the standard deviation shown here, which may introduce non-negligible additional bias. See more details in Sect. 4.3.

|  | Pollen type | Depolarization ratio at 532 nm | Depolarization ratio at 355 nm |
|---|---|---|---|
| This study, Finland | Silver birch | $0.24 \pm 0.01$ | 0.17 |
| (in the atmosphere) | Scots pine | $0.36 \pm 0.01$ | 0.30 |
| Cao et al. (2010), Canada | Paper birch | $0.33 \pm 0.004$ | $0.08 \pm 0.008$ |
| (in an aerosol chamber) | Virginia pine | $0.41 \pm 0.006$ | $0.20 \pm 0.013$ |

"

And in the conclusion we modified as:
"

This algorithm was first tested and validated through a simulator of synthetic lidar profiles (including a direct model and an inverse model modules). Mathematically, the depolarization ratio for pure pollen can be calculated using the equations given in Sect. 3, if other variables are known or can be assumed. We have developed a retrieval method to estimate the pollen depolarization ratio, which was applied to the lidar observations. The depolarization ratio at 532 nm of pure pollen particles was assessed, resulting to $0.24 \pm 0.01$ and $0.36 \pm 0.01$ for birch and pine pollen, respectively. The uncertainty on assumed backscatter-related Ångström exponent of pure pollen will introduce non-negligible bias in addition as discussed in Sect. 4.3.1.

"

**Technical remarks**

**- Affiliations: "P.O. Box 1627, 5 70211" – seems not necessary and isn't provided for the other institutes**
The correction has been done.

**- P1,L11 / P2,L32: depolarization ratio values/value**
The correction has been done.

**- P3,L17: volume linear depolarization ratio (VDR) and particle linear depolarization ratio (PDR)**
The correction has been done.

**- P4,L18: spoken communication – with whom? Please acknowledge the name of the person**
Thank you for your comment. We have added such information as:
"

*B. pubescens* pollen grains are 18-24 × 22-28 µm in size (Nilsson et al., 1977) and *B. pendula* (Silver birch) pollen grains are more or less of the same size (spoken communication with Sanna Pätsi from Aerobiology, University of Turku).

"

**- P6,L10: non-depolarizing aerosol – the received light is depolarized, but the aerosol is depolarizing, please change it throughout the manuscript**
Thank you for pointing it out, we have modified it throughout the manuscript.

**- P6,L12+L30: this type of indices should not be written in italic – please change it throughout the manuscript**
We have changed these indices to non-italic.

**- P6,L21: "thus six pollen backscattering are simulated." – backscatter coefficients or backscatter coefficient profiles (similar P12,L8)**
Thank you for pointing it out, the correction has been done.

**- P9,L8/9: It would be a good idea to begin a new paragraph with line 9**
We agree. Actually it was a new paragraph, but it was not shown with the presented format.

**- Fig. 1, caption of y-axis: [no m-3] – it is -3**
**- Fig. 3a, caption of y-axis: LR 532 [sr] – unit is missing**
Thank you for pointing them out. The corrections have been done.

---

## Author Comment (AC5) · 25 Oct 2020

**Response to Referees – General modifications**

We would like to thank all reviewers for carefully reading the manuscript and providing useful suggestions to improve the paper. Following the comments from 3 reviewers, we made the following modifications (in red) in general for the manuscript:

**1. Presentation of the method and the structure of manuscript**

We have simplified the presentation of method, and separated the presentation of the methodology from the presentation of the results. The structure of revised manuscript will be:

"
    1 Introduction
    2 Site and instruments
    3 Methodology – a synthetic simulator
        3.1 Direct model – generation of synthetic optical profiles
        3.2 Inverse model – retrieval of depolarization ratio
        3.3 Uncertainty study
    4 Results
        4.1 Pollen grain and intense pollination period
        4.2 Optical properties of pollen layer
            4.2.1 Pollen layer
            4.2.2 Lidar-derived optical properties
        4.3 Estimation of optical properties for pure pollen from lidar observations
            4.3.1 Pollen optical properties at 532 nm
            4.3.2 Pollen optical properties at 1064 nm and 355 nm
    5 Summary and conclusions
"

The number of figures and tables are changed correspondingly.

**2. Manuscript title**

We have changed the title, because the title was a bit misleading as the reader might expect observations from an aircraft. We have also removed "in Finland" as the presented method can also be applied to other sites. The new title is:

"

    Optical characterization of pure pollen types using a multi-wavelength Raman polarization lidar
"

**3. Equations, abbreviation and data format**

We have changed PBD to $\chi_{pollen}$ for the whole manuscript.

We have modified the equations in section 3 (Methodology – a synthetic simulator) to make the presentation clearer.

We also added 2 sections of equation calculations in the supplement for:
    1    Particle linear depolarization ratio calculation (Eq.3 in the manuscript)
    2    Relationship of $\text{Å}_{particle}$ and $\chi_{pollen}$ (Eq.5 in the manuscript)

We changed the format of depolarization ratio value from xx% to 0.xx for the whole manuscript.